# On Multi-objective Policy Optimization as a Tool for Reinforcement Learning: Case Studies in Offline RL and Finetuning

## Abstract

Many advances that have improved the robustness and efficiency of deep reinforcement learning (RL) algorithms can, in one way or another, be understood as introducing additional objectives or constraints in the policy optimization step. This includes ideas as far ranging as exploration bonuses, entropy regularization, and regularization toward teachers or data priors. Often, the task reward and auxiliary objectives are in conflict, and in this paper we argue that this makes it natural to treat these cases as instances of multi-objective (MO) optimization problems. We demonstrate how this perspective allows us to develop novel and more effective RL algorithms. In particular, we focus on offline RL and finetuning as case studies, and show that existing approaches can be understood as MO algorithms relying on linear scalarization. We hypothesize that replacing linear scalarization with a better algorithm can improve performance. We introduce Distillation of a Mixture of Experts (DiME), a new MORL algorithm that outperforms linear scalarization and can be applied to these non-standard MO problems. We demonstrate that for offline RL, DiME leads to a simple new algorithm that outperforms state-of-the-art. For finetuning, we derive new algorithms that learn to outperform the teacher policy.

## 1 Introduction

Deep reinforcement learning (RL) algorithms have solved a number of challenging problems, including in games (Mnih et al., 2015; Silver et al., 2016), simulated continuous control (Heess et al., 2017; Peng et al., 2018), and robotics (OpenAI et al., 2018). The standard RL setting appeals through its simplicity: an agent acts in the environment and can discover complex solutions simply by maximizing cumulative discounted reward. In practice, however, the situation is often more complicated. For instance, without a carefully crafted reward function or sophisticated exploration strategy, learning may require hundreds of millions of environment interactions, or may not be possible at all.

A number of strategies have been developed to mitigate the shortcomings of the pure RL paradigm. These include strategies that regularize the final solution, for instance by maximizing auxiliary rewards (Jaderberg et al., 2017) or the entropy of the policy (Mnih et al., 2016; Haarnoja et al., 2018). Others facilitate exploration, for instance via demonstrations (Brys et al., 2015), shaping rewards (Ng et al., 1999), exploration bonuses (Bellemare et al., 2016; Pathak et al., 2017), or guidance from a pre-trained teacher policy, known as policy finetuning (Xie et al., 2021). Strategies for reusing old environment interactions can in the extreme case enable agents to learn solely from a fixed dataset, known as offline RL (Fu et al., 2020; Gulcehre et al., 2020). In this case regularization is needed to ensure that the learned policy is well supported by the data distribution.

These approaches may seem quite different at first glance, but they can all be understood as introducing additional objectives, or constraints, in the policy optimization step. Since reward maximization and the additional objective(s) are usually in conflict, these algorithmic approaches can also be understood as solving a multi-objective RL (MORL) problem. In practice, the solution to this MORL problem is usually implemented via a form of linear scalarization (LS), i.e., by taking a weighted sum of the objectives. However, LS suffers from a number of limitations: it is sensitive to the scale of the objectives (Abdolmaleki et al., 2020) and is limited in the solutions it can find (Das & Dennis, 1997).

*Our key insight is that an interpretation of algorithmic approaches, such as the above, in terms of MORL provides us with novel tools to solve fundamental challenges in RL.* To demonstrate this, we focus on offline RL and finetuning as case studies. We show recent offline RL algorithms (e.g., CRR (Wang et al., 2020), AWAC (Nair et al., 2020)) can be understood as using LS to solve the MO problem. We hypothesize that by replacing LS with a more appropriate approach, we can obtain better-performing algorithms. We thus propose a new MORL approach, called Distillation of a Mixture of Experts (DiME), that can replace LS in this setting. We derive novel algorithms for finetuning and offline RL, based on DiME. In offline RL, this new algorithm outperforms state-of-the-art (Sec. 6).

Our main contributions are as follows:

- We propose that MORL can be used as a tool for tackling fundamental challenges in RL. We show that existing algorithms (e.g., for offline RL) rely on LS to solve the MO problem. Based on this insight, we explore replacing LS with a new, improved MO algorithm.
- By replacing LS, we derive a novel offline RL algorithm from the multi-objective perspective (Sec. 5.2), that is simple, intuitive and outperforms the state-of-the-art (Sec. 6.2).
- In a similar manner, we also derive new finetuning algorithms (Sec. 5.1), and show that these algorithms can train policies that exceed the teacher's performance (Sec. 6.3).
- To replace LS, we introduce a new MORL approach, DiME. We support DiME by theoretical derivations (Sec. 4.1) and show it outperforms LS on standard MORL tasks (Sec. 6.1).

## 2 RELATED WORK

**Offline RL and Finetuning.** Our case studies in this work will involve two related learning problems: offline (or batch) RL (Levine et al., 2020; Lange et al., 2012) and policy finetuning (Xie et al., 2021). Both leverage existing information, either in the form of a dataset or a teacher policy. In this work, we use proximity to the dataset or policy as an additional objective in a MORL problem.

In offline RL, the goal is to learn a policy from a given, fixed dataset of transitions. While classical off-policy RL algorithms can be used, they perform worse than bespoke algorithms (Fujimoto et al., 2019). This is likely due to the extrapolation errors incurred when learning a Q-function—naïve policy optimization could optimistically exploit such errors. Therefore, many existing approaches attempt to constrain the policy optimization to stay close to the samples present in the available dataset (Peng et al., 2019; Nair et al., 2020; Siegel et al., 2020; Wang et al., 2020). Finetuning seeks to improve upon an expert (or teacher) policy, which can be specified either nonparametrically via a dataset of transitions or as a parametric model. In contrast to offline RL, finetuning happens in an online setting, where the agent is able to interact with the environment. Existing approaches for finetuning minimize the cross-entropy between the teacher and agent policies (Schmitt et al., 2018), based on distillation or mimicking (Rusu et al., 2015; Parisotto et al., 2015).

**Multi-Objective RL.** MORL algorithms train policies for environments with multiple sources of reward. There are single-policy and multi-policy approaches. The former finds a policy that is optimal for a single preference trade-off across rewards. One common technique is to use the trade-off to combine the rewards into a single scalar reward, and then use standard RL to optimize this (Roijers et al., 2013). Typically linear scalarization is used, but it is sensitive to reward scales and cannot always find optimal solutions (Das & Dennis, 1997). Non-linear scalarizations exist (Van Moffaert et al., 2013; Golovin & Zhang, 2020), but are hard to combine with value-based RL. Instead of scalarization, recent work on MO-MPO (Abdolmaleki et al., 2020) relies on constrained optimization.

Multi-policy approaches seek to find a set of policies that cover the Pareto front. Various techniques exist, for instance repeatedly calling a single-policy approach with strategically-chosen trade-off settings (Roijers et al., 2014; Mossalam et al., 2016; Zuluaga et al., 2016), simultaneously learning a set of policies by using a multi-objective variant of Q-learning (Moffaert & Nowé, 2014; Reymond & Nowé, 2019; Yang et al., 2019), learning a manifold of policies in parameter space (Parisi et al., 2016; 2017), or combining single-policy approaches with an overarching objective (Xu et al., 2020).

In contrast, here we consider objectives that are not only environment rewards, but also learning- or regularization-focused such as staying close to a behavioral prior. State-of-the-art MORL approaches, including MO-MPO, cannot be readily applied. We thus introduce a new algorithm, DiME, which is on-par with state-of-the-art and can handle these non-standard objectives (Sec. 4). We use DiME as a vehicle to demonstrate the benefits of taking a multi-objective perspective toward challenges in RL.

| Algorithm | Step 1: Improvement | Step 2: Projection |
|---|---|---|
| LS | $q(a\mid s) \propto \pi_i(a\mid s) \exp\left(\frac{\sum_k \alpha_k Q_k(s,a)}{\eta}\right)$ | $\arg\min_\theta \mathbb{E}_{s\sim\mu}\left[D_{\mathrm{KL}}\big(q(\cdot\mid s)\|\pi_\theta(\cdot\mid s)\big)\right]$ |
| MO-MPO | $q_k(a\mid s) \propto \pi_i(a\mid s) \exp\left(\frac{Q_k(s,a)}{\eta_k(\alpha_k)}\right)$ | $\arg\min_\theta \mathbb{E}_{s\sim\mu}\left[\sum_k D_{\mathrm{KL}}\big(q_k(\cdot\mid s)\|\pi_\theta(\cdot\mid s)\big)\right]$ |
| DiME | $q_k(a\mid s) \propto \pi_i(a\mid s) \exp\left(\frac{Q_k(s,a)}{\eta_k}\right)$ | $\arg\min_\theta \mathbb{E}_{s\sim\mu}\left[\sum_k \alpha_k D_{\mathrm{KL}}\big(q_k(\cdot\mid s)\|\pi_\theta(\cdot\mid s)\big)\right]$ |

Table 1: Comparison of policy improvement step for existing approaches versus DiME. Adjustments to the single-objective setting to accommodate multiple objectives are highlighted in red.

# 3 BACKGROUND

We consider the RL problem defined by a Markov Decision Process (MDP). An MDP consists of states $s \in \mathcal{S}$, actions $a \in \mathcal{A}$, an initial state distribution $p(s_0)$, transition probabilities $p(s_{t+1}|s_t, a_t)$, a reward function $r(s, a) \in \mathbb{R}$, and a discount $\gamma \in [0, 1)$. A policy $\pi_\theta(a|s)$ is a state-conditional distribution over actions, parametrized by $\theta$. Together with the transition probabilities, this gives rise to a state visitation distribution $\mu(s)$. The action-value function is the expected return from choosing action $a$ in state $s$ and then following policy $\pi$: $Q^\pi(s, a) = \mathbb{E}_\pi[\sum_{t=0}^\infty \gamma^t r(s_t, a_t)|s_0 = s, a_0 = a]$. This can also be written as $Q^\pi(s, a) = \mathbb{E}_{s'\sim p}[r(s, a) + \gamma V^\pi(s')]$, where $V^\pi(s) = \mathbb{E}_{a\sim\pi}[Q^\pi(s, a)]$.

In this paper, both the LS and DiME approaches to multi-objective RL follow a policy iteration scheme, repeatedly alternating between policy evaluation and policy improvement. In the policy evaluation step, at iteration $i$ the current policy $\pi_{\theta_i}$ is evaluated by training a separate value function $Q_k^i$ for every objective $k$. This requires bootstrapping[1] according to Q-decomposition (Russell & Zimdars, 2003). Any policy evaluation algorithm can be used to learn these Q-functions. In the policy improvement step, given the previous policy $\pi_i$ (short for $\pi_{\theta_i}$ when clear from context) and associated Q-functions $\{Q_k\}_{k=1}^K$, the aim is to improve the policy for a given state visitation distribution $\mu$.[2]

## 3.1 POLICY IMPROVEMENT: SINGLE OBJECTIVE

DiME introduces a new way to perform policy improvement with respect to multiple objectives, so we will now describe the policy improvement step, first for the single-objective setting. Policy improvement can itself be decomposed into the following two alternating steps (Ghosh et al., 2020).

**Improvement.** This step finds a (nonparametric) policy $q$ that improves with respect to the Q-function, while staying close to the current iterate $\pi_i$, leading to the Lagrangian objective function

$$F(q; \theta_i) = \mathbb{E}_{\substack{s\sim\mu \\ a\sim q}}\Big[Q(s, a)\Big] - \eta\, \mathbb{E}_{s\sim\mu}\Big[D_{\mathrm{KL}}\big(q(\cdot|s)\|\pi_i(\cdot|s)\big)\Big], \tag{1}$$

with the known analytic solution $q(a|s) \propto \pi_i(a|s) \exp(Q(s, a)/\eta)$ (Peters et al., 2010).

**Projection.** This step projects the improved policy to the space of parametric policies by maximizing the following (which simplifies to policy gradient if one uses a parametric $q$ in the improvement step):

$$J(\theta) = -\mathbb{E}_{s\sim\mu}\, D_{\mathrm{KL}}\big(q(\cdot|s)\,\|\,\pi_\theta(\cdot|s)\big). \tag{2}$$

## 3.2 POLICY IMPROVEMENT: MULTIPLE OBJECTIVES

Suppose the policy must improve with respect to $K$ objectives instead. In multi-objective problems, a solution is *Pareto-optimal* if there is no other solution with better performance for one objective without worse performance for another. The set of all Pareto-optimal solutions is the *Pareto front*. To find a specific Pareto-optimal policy, let $\underline{\alpha} = \{\alpha_k\}_{k=1}^K$ define a preference trade-off across objectives, such that all $\alpha_k \geq 0$. We can further enforce $\sum_k \alpha_k = 1$. A larger $\alpha_k$ means the $k$-th objective is more important, and should lead to increased influence of objective $k$ on the policy optimization.

**Linear Scalarization.** Here the trade-off is used to produce a single scalar reward via a weighted sum of the objectives. This leads to the nonparametric solution for $q$ in Table 1, while the projection step is the same as for a single objective. A drawback of LS is that not only the trade-offs, but also the scales of the $Q_k$, influence the contribution of each objective.

---

[1] Not all objectives are necessarily evaluated via bootstrapping, but we denote all as $Q_k$ to lighten notation.
[2] We will drop the superscript $i$ on $Q_k^i$, always assuming the latest iterate of the policy evaluation algorithm.

**MO-MPO.** In contrast, Abdolmaleki et al. (2020) optimizes (1) separately for each objective, to obtain $K$ improved policies $\{q_k\}_{k=1}^K$. The trade-offs $\{\alpha_k\}$ specify the constraint threshold on the KL-divergence between $q_k$ and $\pi_i$. As seen in Table 1, this leads to nonparametric solutions with temperatures $\eta_k$, that are each a solution to a convex optimization problem involving $\alpha_k$.

A main advantage of MO-MPO is that it combines objectives in a *scale-invariant* way[3], unlike LS. However, because the trade-offs in MO-MPO define KL-constraints, the improvement operator used for $q_k$ must be able to meet this constraint exactly. This means MO-MPO cannot be easily applied to non-standard settings like offline RL, as we will discuss later in Sec. 5.2. Thus we propose DiME, which can be applied to non-standard settings and maintains the benefit of scale-invariance.

## 4 DISTILLED MIXTURE OF EXPERTS

Like MO-MPO, DiME computes a separate $q_k$ per objective. As seen in Table 1, the nonparametric solution is almost identical, with the crucial exception that the temperature $\eta_k$ no longer depends on $\alpha_k$. Unlike LS and MO-MPO, DiME only incorporates the trade-offs in the projection step objective

$$J_{\text{DiME}}(\theta) = -\mathbb{E}_{s\sim\mu} \sum_{k=1}^K \alpha_k D_{\text{KL}}\Big(q_k(\cdot|s) \,\|\, \pi_\theta(\cdot|s)\Big), \tag{3}$$

so any improvement operator can be used to obtain $q_k$. This allows us to apply DiME to offline RL and other non-standard settings. In addition, the trade-offs in DiME are more interpretable than those in MO-MPO, since they are weights on local experts $q_k$, rather than KL-divergence constraints. Appendix B describes DiME in more detail, including how the temperature $\eta_k$ is computed.

However, we may not *a priori* know what the preference trade-off across objectives should be. In this case we can extend DiME to learn a conditional policy, thereby distilling the entire Pareto front of optimal policies into a single $\pi_\theta(\cdot|s,\underline{\alpha})$. We train this policy by augmenting states with normalized trade-offs $\underline{\alpha}$ sampled from some distribution $\nu$, and then optimizing the following objective:

$$J_{\text{DiME}}^\alpha(\theta) = -\mathbb{E}_{\substack{s\sim\mu \\ \underline{\alpha}\sim\nu}} \sum_{k=1}^K \underline{\alpha}_k D_{\text{KL}}\Big(q_k(\cdot|s,\underline{\alpha}) \,\|\, \pi_\theta(\cdot|s,\underline{\alpha})\Big), \tag{4}$$

where $q_k(a|s,\underline{\alpha}) \propto \pi_i(a|s,\underline{\alpha})\exp(Q_k(s,a,\underline{\alpha})/\eta_k)$. Note that in order to evaluate this policy, we must also learn an $\underline{\alpha}$-conditioned value function $Q_k(s,a,\underline{\alpha})$. Appendix A contains more details.

After training an $\alpha$-conditioned policy, one can use $Q_k(\alpha) = \mathbb{E}_{s\sim\mu} \int \pi(a|s,\alpha)Q_k(s,a,\alpha)\,\mathrm{d}a$ fully offline to pick the best trade-off $\alpha$ with respect to objective $k$, using any hyperparameter optimization (HPO) algorithm, including a simple search. If the learned $Q_k$ are inaccurate, which is often true in offline RL, we can instead deploy the policy to search for the best trade-off. This is cheap; in our case studies $\alpha$ is between 0 and 1. In this paper we aim to learn high-quality $\alpha$-conditioned policies, which we will show is possible using DiME. Investigating other ways of picking $\alpha$ is out of this scope.

### 4.1 MULTI-OBJECTIVE RL AS PROBABILISTIC INFERENCE

We will now derive LS and DiME from the *RL as inference* perspective for MORL, showing that both optimize for the same high-level problem, but differ in their choice of variational distribution. Consider a binary random variable $R$ indicating a policy improvement event: $R = 1$ indicates the policy has improved, while $R = 0$ indicates that it has not. Policy optimization then corresponds to maximizing the marginal likelihood $\mathbb{E}_\mu \log p_\theta(R|s)$ with respect to $\theta$. Employing the expectation-maximization (EM) algorithm and assuming a conditional probability $p(R|s,a) \propto \exp(Q(s,a)/\eta)$, one can recover many algorithms in the literature (Levine, 2018).

Let us instead define a separate indicator $R_k$ for each objective, with similarly defined conditional probabilities, and consider the following weighted log-likelihood objective:

$$F(\theta) = \mathbb{E}_\mu \log \prod_{k=1}^K p_\theta(R_k|s)^{\alpha_k} = \sum_{k=1}^K \alpha_k \mathbb{E}_\mu \log p_\theta(R_k|s), \tag{5}$$

---

[3]if the improvement step is scale-invariant, which is true for the known nonparametric solution if we optimize the temperature $\eta$ to meet a constraint on the KL-divergence, as is done in e.g. MPO (Abdolmaleki et al., 2018).

where $\mathbb{E}_\mu$ is a shorthand for $\mathbb{E}_{s\sim\mu}$ and $\alpha \in \mathbb{R}_+^K$ specify a preference trade-off across the $K$ objectives. This generalization has a few nice properties. First, when all objectives are equally preferred, the $\alpha_k$ are all equal and it reduces to a straightforward probabilistic factorization of independent events: $p(\{R_k\}_{k=1}^K) = \prod_k p(R_k)$. Second, any vanishing $\alpha_k$ leads to an objective being ignored.

**Linear Scalarization.** To apply EM, we could introduce a variational distribution $q(a|s)$ and use it to decompose and lower-bound (5). This choice of variational distribution results in LS: the E-step corresponds to (1) with $Q(s, a)$ replaced by $\sum_k \alpha_k Q_k(s, a)$, and the M-step is identical to (2).

**DiME.** We could instead introduce a *separate* variational distribution $q_k$ for each $\log p_\theta(R_k|s)$, which results in DiME. The E-step now corresponds to $K$ independent copies of the improvement step in (1), each with its own $Q_k(s, a)$. The M-step is the projection step introduced in (3).

This reveals that both standard LS and DiME optimize the same information-theoretic objective (see Appendix B.1 for a complete derivation). Both employ EM, but differ in their choice of variational decomposition. We hypothesize that fitting a separate variational distribution per objective allows DiME to find a tighter variational lower bound, which may lead to better optimization.

# 5 CASE STUDIES

Our primary aim in this work is to showcase MORL as a tool for tackling fundamental challenges in deep RL. We therefore demonstrate its benefits in two active areas of research: finetuning and offline RL. Finetuning is concerned with improving upon an expert (or teacher) policy, whether it is available in the form of a parametric model or a dataset of transitions. Meanwhile in offline RL, one must learn a policy from a given fixed dataset of transitions. In contrast to offline RL, finetuning happens in an online fashion so the agent interacts with the environment to collect additional experience.

These two settings share a fundamental similarity—they both use behavioral priors to make learning more efficient. Existing algorithms in both settings constrain the policy optimization, either explicitly or implicitly, to stay close to the given priors (Levine et al., 2020; Monier et al., 2020; Schmitt et al., 2018). Instead, we propose to explicitly formulate these in the MORL framework with two objectives:

1. one for the task return via the task Q-function $Q(s, a)$ learned via bootstrapping,
2. and one for staying close to the behavioral prior via the log-density[4] ratio $\log \frac{\pi_b(a|s)}{\pi_i(a|s)}$,

where $\pi_i$ is the current policy and $\pi_b$ is the behavioral prior. When we have direct access to the density $\pi_b$, we can evaluate the log-density ratio pointwise. However, in many real-world applications, we only have access to samples from $\pi_b$ via a dataset of historical interactions. In the following subsections, we dive into each of these cases, deriving the corresponding policy losses for both LS and DiME. We use a scalar $\alpha \in [0, 1]$ to specify the preference trade-off between the two objectives.

## 5.1 FINETUNING

Equipped with the MORL perspective, we introduce two novel finetuning algorithms by substituting the objectives above in the LS and DiME rows of Table 1, yielding the policy optimization functions:[5]

$$J_{\text{LS}}(\theta) = \mathbb{E}_{\substack{s\sim\mu \\ a\sim\pi_i}} \left[ \frac{1}{Z_0(s)} \exp\left( (1-\alpha)\frac{Q(s,a)}{\eta} + \frac{\alpha}{\eta}\log\frac{\pi_b(a|s)}{\pi_i(a|s)} \right) \log \pi_\theta(a|s) \right] \text{ and} \quad (6)$$

$$J_{\text{DiME}}(\theta) = \mathbb{E}_{\substack{s\sim\mu \\ a\sim\pi_i}} \left[ \left( \frac{(1-\alpha)}{Z_1(s)} \exp\left( \frac{Q(s,a)}{\eta_1} \right) + \frac{\alpha}{Z_2(s)} \exp\left( \frac{\log\frac{\pi_b(a|s)}{\pi_i(a|s)}}{\eta_2} \right) \right) \log \pi_\theta(a|s) \right]. \quad (7)$$

All functions $Z_*(s)$ normalize the product of $\pi_i$ and the corresponding tempered and exponentiated objective; though analytically intractable, they can be estimated from sampled actions for each state. Note that both of the equations above are equivalent at the extreme values of $\alpha = 0$ and 1.

**Relation to existing approaches.** We can recover existing algorithms with slight modifications to these policy objectives. In particular, setting $\eta_2 = 1$ in the DiME row of Table 1 yields $q_2 = \pi_b$ and

---

[4]The log-density ratio is typically used as reward for distribution matching (Arenz & Neumann, 2020).
[5]Here, and in our finetuning experiments, we assume the prior is a parametric model. If the prior is instead a dataset of transitions, then the update rules are identical to the ones derived later for offline RL.

the following policy objective function:

$$(1-\alpha)\,\mathbb{E}_{\substack{s\sim\mu \\ a\sim\pi_i}}\left[\frac{1}{Z_1(s)}\exp\left(\frac{Q(s,a)}{\eta}\right)\log\pi_\theta(a|s)\right]-\alpha\,\mathbb{E}_{s\sim\mu}D_{\mathrm{KL}}\big(\pi_b\|\pi_\theta\big)\,,\quad(8)$$

which is similar to the finetuning approach in (Schmitt et al., 2018). Similarly, taking instead the LS approach to MORL and employing the policy gradient (which uses a parametric $q$ in Sec. 3.1) as the underlying policy optimization, we get:

$$\mathbb{E}_{\substack{s\sim\mu \\ a\sim\pi_\theta}}\left[(1-\alpha)Q(s,a)+\alpha\log\frac{\pi_b(a|s)}{\pi_\theta(a|s)}\right]=(1-\alpha)\mathbb{E}_{\substack{s\sim\mu \\ a\sim\pi_\theta}}Q(s,a)-\alpha\mathbb{E}_{s\sim\mu}D_{\mathrm{KL}}\big(\pi_\theta\|\pi_b\big)\,,\quad(9)$$

which is similar to (Galashov et al., 2019). Both simply add a weighted KL regularization to a standard RL objective, but they differ in the direction of the KL.

## 5.2 OFFLINE RL

Recall that in offline RL, we only have access to the behavioral prior $\pi_b$ via a dataset $\mathcal{D}_{\pi_b}$ of sampled transitions $(s,a,r,s')$; we cannot evaluate its density and thus cannot directly compute the objective function for staying close to the prior. Instead, we must simplify both $J_{\mathrm{LS}}$ and $J_{\mathrm{DiME}}$ from the finetuning subsection so that they only require samples from $\pi_b$ and $\mu$, for which we can rely on $\mathcal{D}_{\pi_b}$.

**Linear Scalarization.** By tying the preference and scale parameters so that $\alpha=\eta$, we can change the expectation over actions in (6) analytically to be with respect to the behavior policy $\pi_b$

$$J_{\mathrm{LS}}(\theta)=\mathbb{E}_{\mathcal{D}_{\pi_b}}\left[\exp\left(\frac{Q(s,a)}{\beta}-\log Z_0\right)\log\pi_\theta(a|s)\right]\,,\quad(10)$$

where $\beta=\alpha/(1-\alpha)$. There are close similarities between this LS-based policy objective and other policy losses used in recent offline RL algorithms. For example, $\beta$ corresponds to the temperature in CRR (Wang et al., 2020) and AWAC (Nair et al., 2020), and we can recover them exactly by replacing the task return objective with the advantage function. As an other example, BEAR (Kumar et al., 2019) and BCQ (Fujimoto et al., 2019) can both be formulated as using LS, but with the policy gradient as their underlying policy optimizer as in (9). However, unlike the objective function in (10), which only requires samples from $\pi_b$, these methods have the limitation that they require modeling $\pi_b$ to evaluate the second term in (9) (Levine et al., 2020; Monier et al., 2020).

**DiME.** Setting $\eta_2=1$ and renaming $\eta_1=\eta$ in (7), we obtain

$$J_{\mathrm{DiME}}(\theta)=(1-\alpha)\,\mathbb{E}_{\substack{s\sim\mu \\ a\sim\pi_i}}\left[\frac{1}{Z_1(s)}\exp\left(\frac{Q(s,a)}{\eta}\right)\log\pi_\theta(a|s)\right]+\alpha\,\mathbb{E}_{\mathcal{D}_{\pi_b}}\log\pi_\theta(a|s).\quad(11)$$

Notice that the first term is the loss used by policy optimization methods such as REPS (Peters et al., 2010) and MPO (Abdolmaleki et al., 2018), and the second term is exactly a behavioral cloning loss. We call this new algorithm DiME (BC). This is a simple way of adapting online algorithms for offline RL, while being theoretically grounded in a multi-objective perspective. Intuitively, since both terms are independent of the scale of the Q-values, optimizing the weighted sum is a reasonable thing to do. This is in contrast to recent concurrent work that adds a behavioral cloning loss to a policy gradient loss, which depends on the scale of the Q-values (Fujimoto & Gu, 2021).

**Remark on drawbacks of LS approaches.** First, the preference $\alpha$ and scale $\eta$ can no longer be set independently; they are coupled together in the temperature $\beta$. This means we cannot efficiently train a trade-off-conditioned policy using the LS-based loss in (10), because the normalization constant $Z_0=\int_{(s,a)\sim\mathcal{D}_{\pi_b}}\exp(Q(s,a)/\beta)$ depends on the choice of the trade-off—this would require estimating (10) via samples separately per trade-off. Second, since $J_{\mathrm{LS}}$ only involves an expectation with respect to the expert behavior policy, the policy cannot evaluate and exploit the $Q$-value estimates for actions beyond those taken by the expert. One could argue that this is by design, in order to avoid fitting extrapolation errors in the Q-function. However, with $J_{\mathrm{DiME}}$ this is a choice we can make by setting the preference parameter independently (and perhaps adaptively).

**Remark on using other MORL approaches.** Although there exist many MORL algorithms that outperform LS on standard MORL tasks, they typically require that the rewards for all objectives are given. Thus it is not straightforward to apply these algorithms for offline RL, where we cannot directly compute the reward for staying close to the prior. We tried getting around this in order to use MO-MPO for offline RL, but it was very complicated to implement (for details see Appendix B.4).

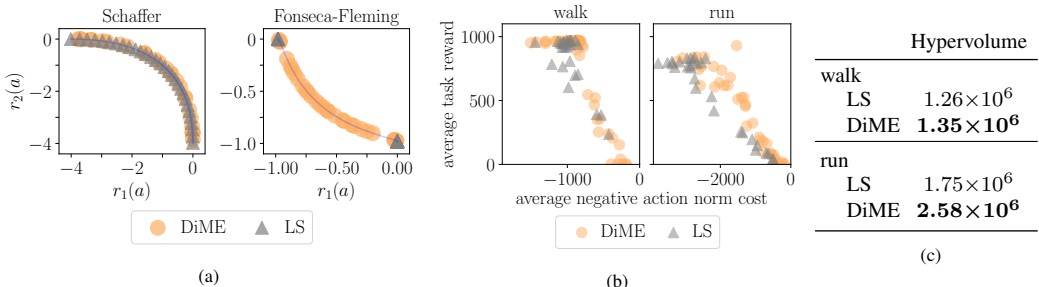

Figure 1: Each point is a solution found for a different tradeoff $\alpha$. (a) DiME can find solutions on a concave Pareto front (right) whereas linear scalarization (LS) cannot. (b,c) For humanoid, DiME finds better solutions and obtains higher hypervolume. Above (i.e., higher task reward) and to the right (i.e., lower cost) is better.

## 6 EXPERIMENTS

We first investigate how DiME performs on standard MORL tasks, compared to LS (Sec. 6.1). Since our focus is on applying MORL for general challenges in RL, we do not consider MORL algorithms that cannot be applied to offline RL. Then we investigate the application of DiME in two case studies, offline RL (Sec. 6.2) and finetuning (Sec. 6.3), where our aim is to evaluate whether replacing LS with DiME leads to better-performing algorithms. Implementation details are in Appendix C.

### 6.1 MULTI-OBJECTIVE RL

**Toy Domain.** Our bandit task has action $a \in \mathbf{R}$ and reward $r(a) \in \mathbf{R}^2$, with the reward defined by either the Schaffer or Fonseca-Fleming function (Okabe et al., 2004). The Pareto front is the set of all points of the function. By varying the trade-off, DiME finds solutions along the entire Pareto front for both functions. However, no matter what the trade-off is, LS only finds solutions at the extremes of Fonseca-Fleming, because this Pareto front is concave (Fig. 1a). This is a fundamental limitation of LS—it cannot find solutions on the concave portions of a Pareto front (Das & Dennis, 1997).

**Humanoid.** We also evaluate on two humanoid tasks from the DeepMind Control Suite (Tassa et al., 2018). For each, the original task reward objective is augmented with an "energy"-expenditure penalty on actions: $-\|a\|_2$. In both tasks, the Pareto front found by DiME is superior to that found by LS, both qualitatively (Fig. 1b) and in terms of hypervolume, which is a standard metric for capturing the overall quality of a Pareto front (Table 1c).

**Additional Experiments.** We also find that DiME is scale-invariant—it obtains similar performance when the energy cost is scaled up by ten times, whereas LS no longer can learn the task. DiME also performs on-par with MO-MPO, which is state-of-the-art. Please refer to Appendix D for details. Recall that DiME can be applied to non-standard settings like offline RL, whereas MO-MPO cannot.

### 6.2 OFFLINE RL

**Baselines.** For offline RL, our main baseline uses LS to trade off between the task return and staying close to the behavioral prior, with the advantage function as the task return objective. This optimizes for $\mathbb{E}_{(s,a)\sim\mathcal{D}_{\pi_b}} \exp\left((1-\alpha)/\alpha A(s,a)\right) \log \pi_\theta(a|s)$. This is the same policy loss as CRR-exp (Wang et al., 2020) and AWAC (Nair et al., 2020), two state-of-the-art offline RL algorithms. Our behavioral cloning (BC) baseline optimizes for $\mathbb{E}_{(s,a)\sim\mathcal{D}_{\pi_b}} \log \pi_\theta(a|s)$, which is LS or DiME with $\alpha = 1$.

**Variations of DiME.** For offline RL, our DiME-based algorithm optimizes for (11), where the second term is a behavior cloning loss; we call this DiME (BC). For this second term, we could instead use an advantage-weighted behavior cloning loss, $\mathbb{E}_{\pi_b(a|s)}[\exp(A(s,a)) \log \pi_\theta(a|s)]$, which is exactly the CRR policy loss with a temperature of 1; we call this DiME (AWBC). Since training a separate policy per trade-off is computationally expensive, we also experiment with training a single trade-off-conditioned policy $\pi_\theta(a|s,\alpha)$, that can optimize for all trade-offs in the range $[0,1]$. We refer to this as DiME multi, and combine it with both BC and AWBC. We choose to train a single policy that can optimize for all trade-offs, rather than learning the best trade-off during training, because it is a well-known problem that policy evaluation is inaccurate in offline RL.

| Task | BRAC | CRR† | MZU | BC | LS (CRR) α=.3 | DiME (BC) α=.45 | DiME (AWBC) α=.5 | LS (CRR) | DiME (BC) | DiME (AWBC) | DiME (BC) multi | DiME (AWBC) multi |
|---|---|---|---|---|---|---|---|---|---|---|---|---|
| cartpole swingup | **869** | 664 | 343 | 380 | 677 | **865** | 832 | **843** | **882** | **881** | **878 ± 1** | **878 ± 1** |
| cheetah run | 539 | 577 | **799** | 376 | 628 | 705 | 759 | **796** | 733 | 759 | **808 ± 7** | **802 ± 9** |
| finger turn hard | 227 | 714 | 405 | 132 | 360 | 741 | 837 | 515 | **885** | **892** | **899 ± 7** | **903 ± 4** |
| fish swim | 222 | 517 | 585 | 493 | 566 | 652 | 665 | 613 | 652 | 669 | **721 ± 4** | **701 ± 6** |
| humanoid run | 10 | 586 | **633** | 390 | 507 | 596 | 582 | **635** | **638** | **660** | **636 ± 3** | **661 ± 2** |
| manip insert ball | 56 | 625 | 557 | 370 | 575 | 554 | 578 | 603 | 652 | 626 | **699 ± 5** | **709 ± 3** |
| manip insert peg | 50 | 387 | **433** | 261 | 409 | 304 | 281 | 410 | 386 | **435** | **432 ± 7** | **436 ± 8** |
| walker stand | 829 | 797 | 760 | 286 | 672 | 783 | 861 | 706 | **955** | **951** | **964 ± 1** | **967 ± 1** |
| walker walk | 786 | 901 | 902 | 361 | 865 | 842 | 796 | 868 | **955** | **957** | **961 ± 1** | **962 ± 1** |
| mean score | 399 | 641 | 602 | 339 | 584 | 671 | 688 | 665 | **749** | **759** | **778** | **780** |

| Task | BRAC | BEAR | AWR | BCQ | CQL | BC | LS (CRR) α=.63 | LS (CRR) | DiME (BC) multi | DiME (AWBC) multi |
|---|---|---|---|---|---|---|---|---|---|---|
| antmaze mean score | 0.23 | 0.23 | 0.22 | 0.25 | 0.48 | 0.15 | 0.78 | **0.89** | 0.78 | **0.88** |
| kitchen mean score | 0.0 | 0.80 | 0.33 | 0.47 | 1.90 | 2.09 | 2.16 | **2.49** | **2.55** | 2.13 |

Table 2: Offline RL results for state-of-the-art approaches (left columns), and our baselines and DiME with either a single trade-off across all tasks (middle) or the best trade-off per task (right). Tasks are from RL Unplugged (above) and D4RL (below). For DiME multi we include standard error bounds across ten random seeds. Numbers within 5% of the best score are in bold. DiME performs on par or better than LS across all tasks. Even when the trade-off is fixed across all tasks (for a more fair comparison with prior work), DiME still outperforms state-of-the-art on the RL Unplugged tasks. For RL Unplugged, the results for BRAC are from Gulcehre et al. (2020); CRR is from Wang et al. (2020), referred to as "CRR exp" in their paper; and MZU (MuZero Unplugged) is from Schrittwieser et al. (2021). For D4RL, the results for the italicized algorithms are from Fu et al. (2020). †Note that the CRR results are for the best checkpoint from training, whereas all other results denote performance after a fixed number of learning steps. Per-task results for D4RL are in Appendix D.

**Evaluation.** We evaluate on tasks from two offline RL benchmarks, RL Unplugged (Gulcehre et al., 2020) and D4RL (Fu et al., 2020), with results in Table 2. For LS, DiME (BC), and DiME (AWBC), we train one policy per trade-off $\alpha$, linearly spaced from 0.05 to 1. For DiME (BC) multi and DiME (AWBC) multi, we train ten policies with different random seeds. We train all policies for one million learner steps. After training, we evaluate the policy in the test-time environment once per trade-off.

From RL Unplugged, we evaluate on the nine Control Suite tasks. For both LS and DiME, the setting of $\alpha$ has a significant impact on performance, and the optimal setting is different per task (Fig. 2). Comparing the best performance obtained for each task (across all trade-offs), all DiME variants outperform LS. In addition, training a single tradeoff-conditioned policy (with DiME multi) is not only more efficient, but also leads to better performance. This may be because conditioning on $\alpha$ regularizes learning. With LS though, we cannot efficiently train $\alpha$-conditioned policies because $\alpha$ controls both the trade-off and scaling. To make a more fair comparison against prior work, we also find the best fixed trade-off across all tasks for LS and DiME (BC/AWBC), and report these results in the middle columns of Table 2. Both DiME variants still outperform existing approaches.

From D4RL, we evaluate on the six Ant Maze tasks and three Franka Kitchen tasks. We only evaluate the DiME multi variants, with one random seed, since they perform best on the Control Suite tasks and have low variance. LS and DiME perform on par, and both outperform state-of-the-art approaches.

### 6.3 Finetuning

Finally, we will evaluate whether our new finetuning algorithms, derived from the multi-objective perspective, can improve upon the performance of a teacher policy. We evaluate both the LS-based and DiME-based algorithm with either a fixed or learned tradeoff. To learn the tradeoff, we update $\alpha$ based on the loss $J(\alpha) = \alpha \left( \mathbb{E}_{s \sim \mu, a \sim \pi_i} Q(s, a) - c \right)$. This is analogous to the update for the Lagrange multiplier in Lagrangian relaxation for constrained optimization. The resulting strategy is intuitive: stay close to the behavioral prior while the expected return is below the threshold $c$, and otherwise optimize for the bootstrapped Q-function. We pick $c$ based on the expected return from fully imitating the prior for the given task (i.e., $\alpha = 1$); these values are given in Appendix C.

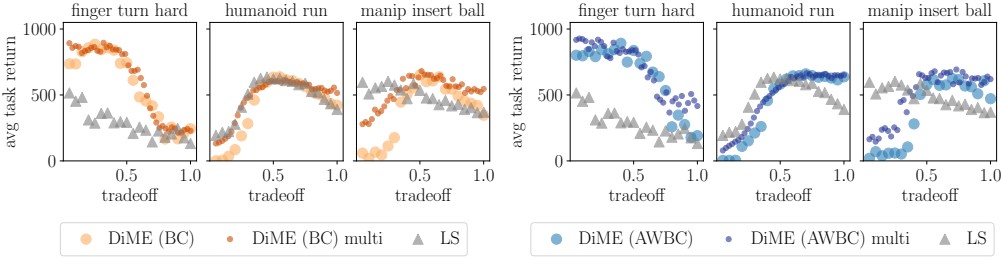

Figure 2: Per-tradeoff performance for a representative subset of tasks; Appendix D contains plots for all fifteen tasks. On some tasks, the best solution found by DiME (any variation) obtains significantly higher task performance than the best found by LS. In addition, DiME multi can train a *single* policy for a range of trade-offs, that performs comparably to learning a separate policy for each tradeoff, after the same number of learning steps.

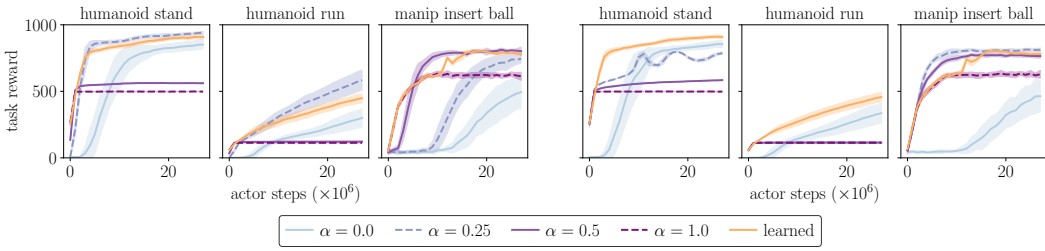

Figure 3: Finetuning learning curves, for DiME (left three) and LS (right three). (Appendix D contains plots for all five tasks.) For both DiME and LS, learning the tradeoff (orange) converges to better performance than fully imitating the teacher (dashed purple), while learning as quickly. Error bars show standard deviation for ten seeds.

We evaluate on the humanoid and manipulator tasks from the DeepMind Control Suite. For the three humanoid tasks, we start with a suboptimal humanoid stand policy as the behavioral prior. For the two manipulator tasks, the behavioral prior is a policy trained for that task.

The choice of fixed tradeoff impacts the result significantly, and the optimal choice depends on the task, so either searching over or learning the tradeoff is necessary. Across all five tasks, for both LS and DiME, the learned tradeoff converges to better performance than fully imitating the prior ($\alpha = 1$) and learns much more quickly than learning from scratch ($\alpha = 0$). This has the advantage of training only a single policy, as opposed to training a separate policy per fixed tradeoff. Adapting the trade-off helps because in finetuning, the agent is able to gather online interactions. An adaptive scheduling for the trade-off allows the agent to initially copy the teacher and then eventually deviate from the teacher to achieve better performance based on its new experiences. In contrast, in the offline RL setting, we needed to use fixed trade-offs because such online interactions are not allowed.

## 7   CONCLUSION AND FUTURE WORK

In this work, we propose using multi-objective optimization as a tool for tackling challenges in RL. We explored the benefits of this in offline RL and finetuning, both of which involve staying close to a behavioral prior. Our derivations revealed that existing approaches optimize for a linear scalarization of two objectives, task return and a log-density ratio. We proposed a new MORL algorithm, DiME, as an alternative to LS. Applying DiME to offline RL leads to state-of-the-art performance.

A key takeaway is the impact of the trade-off $\alpha$ on performance, and the need to either search over or optimize for it. As a first solution we explored either training trade-off-conditioned policies or adapting the trade-off during training; it is worth exploring other approaches in future work.

In offline RL, a limitation of optimizing for the trade-off is that it requires being able to accurately evaluate the policy for multiple trade-offs. We chose the simplest option in our experiments: deploy the fully-trained policy for each trade-off in the test-time environment, and pick the best-performing trade-off. But this may not be feasible in real-world applications, for instance if it is expensive or dangerous to deploy a policy. One option is to use recent offline policy evaluation algorithms to accurately estimate a policy's test-time performance (for all trade-offs) without requiring deployment.

REPRODUCIBILITY STATEMENT

In Appendix C, we provide full implementation details for DiME, our novel algorithms for finetuning and offline RL, and the baselines. We report hyperparameter settings and network architectures in Tables 3 and 4, also in Appendix C. Our code is based on the open-sourced implementations of MO-MPO and MPO in Acme (Hoffman et al., 2020), which is an open-source RL framework. We are in the process of open-sourcing the code.

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

## A  TRAINING TRADE-OFF-CONDITIONED POLICIES

In this section we provide details on training a trade-off-conditioned policy $\pi_\theta(a|s, \alpha)$ conditioned on preference parameters $\alpha \sim \nu(\alpha)$. (Note that here $\alpha$ corresponds exactly to $\underline{\alpha}$ from the main body of this manuscript; we had to distinguish it from the single scalar that appears in our case studies, but we drop the underline here to lighten notation.) Conditioning the policy on the trade-off $\alpha$ requires that the Q-functions are also trade-off-conditioned. We also use hindsight relabeling of trade-offs to perform off-policy learning.

**Preference-conditioned policy evaluation.**  We evaluate the current policy $\pi_i(a|s, \alpha)$ via off-policy learning of per-objective Q-functions $Q_k^i(s, a, \alpha)$. When training the Q-network, every sampled transition $(s, a, \{r_k\}_{k=1}^K, s')$ is augmented such that the states include a sampled trade-off parameter $\alpha \sim \nu$, resulting in transitions $([s, \alpha], a, \{r_k\}_{k=1}^K, [s', \alpha])$. From here, any critic learning algorithm can be used; we use distributional Q-learning (Bellemare et al., 2017; Barth-Maron et al., 2018). See Section C for more details.

**Preference-conditioned policy improvement.**  The policy improvement step produces a trade-off-conditioned policy $\pi_{i+1}(a|s, \alpha)$ that improves upon its predecessor $\pi_i$.

*Improvement:* To find per-objective improved action distributions $q_k(a|s, \alpha)$, we optimize the following policy optimization problem for each objective:

$$\max_{q_k} \quad \mathbb{E}_{\substack{s \sim \mu \\ \alpha \sim \nu}} \left[ \mathbb{E}_{a \sim q_k(\cdot|s, \alpha)} Q_k^i(s, a, \alpha) \right] \tag{12}$$

$$\text{s.t.} \quad \mathbb{E}_{\substack{s \sim \mu \\ \alpha \sim \nu}} D_{\mathrm{KL}}(q_k(a|s, \alpha) \| \pi_i(a|s, \alpha)) \leq \epsilon_k. \tag{13}$$

Similarly, from here we can follow steps analogous to those in Section B.1, but with trade-off-conditioned policies and Q-functions. This means that this set of trade-off-conditioned problems have solutions that are analogous to (38) for each objective.

*Projection:* After obtaining per-objective improved policies $q_k$, we can use supervised learning to distill these distributions into a new parameterized policy:

$$\max_{\theta} \quad \mathbb{E}_{\substack{s \sim \mu \\ \alpha \sim \nu}} \sum_{k=1}^K \alpha_k D_{\mathrm{KL}}\Big( q_k(\cdot|s, \alpha) \, \| \, \pi_\theta(\cdot|s, \alpha) \Big) \tag{14}$$

$$\text{s.t.} \quad \mathbb{E}_{\substack{s \sim \mu \\ \alpha \sim \nu}} \big[ D_{\mathrm{KL}}(\pi_i(\cdot|s, \alpha) \| \pi_\theta(\cdot|s, \alpha)) \big] \leq \beta,$$

subject to a trust region with bound $\beta > 0$ for more stable learning. To solve this optimization, we use Lagrangian relaxation as described in Abdolmaleki et al. (2018; 2020).

**In practice.**  To approximate the expectations over the state and trade-off distributions, we draw states from a dataset (or replay buffer in online RL) and augment each with a sampled trade-off $\alpha \sim \nu$. To approximate the integrals over actions $a$, for each $(s, \alpha)$ pair we sample $N$ actions from the current policy $\pi_i(a|s, \alpha)$.

## B  ALGORITHMIC DETAILS AND DERIVATIONS

### B.1  MULTI-OBJECTIVE RL AS PROBABILISTIC INFERENCE

In this section we present further details of our derivation, which unifies both linear scalarization and DiME, relying on the *RL as inference* framework. We begin by recalling some of the motivation behind the objective.

Consider a set of binary random variables $R_k$ indicating a policy improvement event with respect to objective $k$; $R_k = 1$ indicates that our policy has improved for objective $k$, while $R_k = 0$ indicates that it has not. We seek a policy that improves with respect to *all* objectives given some preferences over objectives. Concretely, we seek policy parameters $\theta$ that maximize the following marginal likelihood, given preference trade-off coefficients $\{\alpha_k\}$:

$$\max_{\theta} \mathbb{E}_\mu \log p_\theta(\{R_k = 1\}_{k=1}^K | \{\alpha_k\}_{k=1}^K). \tag{15}$$

Assuming independence between improvement events $R_k$ we use the following model to enforce trade-offs between objectives, i.e,

$$F(\theta) = \mathbb{E}_\mu \log \prod_{k=1}^{K} p_\theta(R_k|s)^{\alpha_k} = \sum_{k=1}^{K} \alpha_k \mathbb{E}_\mu \log p_\theta(R_k|s). \tag{16}$$

This generalization has a few nice properties. First, when all objectives are equally preferred, the $\alpha_k$ are all equal and it reduces to a straightforward probabilistic factorization of independent events: $p(\{R_k\}_{k=1}^{K}) = \prod_k p(R_k)$. Second, any vanishing $\alpha_k$ leads to an objective being ignored.

Consider the intractable marginal likelihood $L$ that we wish to maximize, then introduce variational distributions $q_k$ and expand as follows

$$F(\theta) = \sum_{k=1}^{K} \alpha_k \mathbb{E}_\mu \log p_\theta(R_k|s) \tag{17}$$

$$= \sum_{k=1}^{K} \alpha_k \mathbb{E}_\mu \mathbb{E}_{q_k} \log p_\theta(R_k|s) \tag{18}$$

$$= \sum_{k=1}^{K} \alpha_k \mathbb{E}_\mu \mathbb{E}_{q_k} \log \frac{p_\theta(R_k, a|s)}{p_\theta(a|R_k, s)} \tag{19}$$

$$= \sum_{k=1}^{K} \alpha_k \mathbb{E}_\mu \mathbb{E}_{q_k} \log \frac{q_k(a|s)\, p_\theta(R_k, a|s)}{p_\theta(a|R_k, s)\, q_k(a|s)} \tag{20}$$

$$= \sum_{k=1}^{K} \alpha_k \mathbb{E}_\mu \left[ D_{\mathrm{KL}}(q_k(a|s) \,\|\, p_\theta(a|R_k, s)) - D_{\mathrm{KL}}(q_k(a|s) \,\|\, p_\theta(R_k, a|s)) \right] \tag{21}$$

$$= \sum_{k=1}^{K} \alpha_k \mathbb{E}_\mu \left[ D_{\mathrm{KL}}(q_k(a|s) \,\|\, p_\theta(a|R_k, s)) - D_{\mathrm{KL}}(q_k(a|s) \,\|\, p(R_k|s, a)\pi_\theta(a|s)) \right] \tag{22}$$

Let us denote the second term as $F$, defined as follows

$$L(\theta; \{q_k\}) := -\sum_{k=1}^{K} \alpha_k \mathbb{E}_\mu D_{\mathrm{KL}}(q_k(a|s) \,\|\, p(R_k|s, a)\pi_\theta(a|s)) \tag{23}$$

Then one can rearrange terms in the expansion of the marginal likelihood $F$ to yield

$$F(\theta) - \sum_{k=1}^{K} \alpha_k \mathbb{E}_\mu D_{\mathrm{KL}}(q_k(a|s) \,\|\, p_\theta(a|R_k, s)) = L(\theta; \{q_k\}). \tag{24}$$

Since the KL divergence is a non-negative quantity and the $\alpha_k$ are all positive, this last equation shows that $L$ lower-bounds the quantity we wish to maximize, namely $F$. Whether we call it expectation-maximization or improvement-projection, the steps are as follows:

1. tighten the gap between the lower bound $L$ and $F$ at the current iterate $\theta_i$, by maximizing $L$ with respect to $q_k$ while keeping $\theta = \theta_i$ fixed, then
2. maximize the lower bound $L$ with respect to $\theta$, holding $q_k$ fixed to the solution of step 1.

Formally, these two steps result in the following optimization problems

$$\max_{\{q_k\}} L(\theta_i; \{q_k\}) = \min_{\{q_k\}} \sum_{k=1}^{K} \alpha_k \mathbb{E}_\mu D_{\mathrm{KL}}(q_k(a|s) \,\|\, p(R_k|s, a)\pi_i(a|s)) \tag{25}$$

$$\max_{\theta} L(\theta; \{q_k\}) = \min_{\theta} \sum_{k=1}^{K} \alpha_k \mathbb{E}_\mu D_{\mathrm{KL}}(q_k(a|s) \,\|\, \pi_\theta(a|s)), \tag{26}$$

where in the second step we removed terms that do not depend on $\theta$. These steps are respectively referred to as the E- and M-step in the statistics literature, and improvement and projection in policy optimization. While these two optimization problems seem decoupled from the original goal of maximizing $F$, the connection with the EM algorithm allows us to benefit from the guarantee that iterates $\theta_i$ obtained in this way lead to improvements in $F$.

## B.2 IMPROVEMENT OR E-STEP

In the next sections we detail the two steps in turn, beginning with the improvement E-step, where we first notice that the objective is a weighted sum of $K$ independent terms that can all be minimized separately

$$\arg\min_{\{q_k\}} \sum_{k=1}^{K} \alpha_k \mathbb{E}_\mu D_{\mathrm{KL}}(q_k(a|s) \,\|\, p(R_k|s,a)\pi_i(a|s)) \tag{27}$$

$$\Leftrightarrow \quad \arg\min_{q_k} \mathbb{E}_\mu D_{\mathrm{KL}}(q_k(a|s) \,\|\, p(R_k|s,a)\pi_i(a|s)), \quad \forall k = 1, \dots, K. \tag{28}$$

These separate problems can be solved analytically by making the KL vanish with the following solution

$$q_k(a|s) = \frac{p(R_k|s,a)\pi_i(a|s)}{\int \pi_i(a|s)p(R_k|s,a)\,\mathrm{d}a} \tag{29}$$

This nonparametric distribution reweights the actions based on the improvement likelihood $p(R_k|s,a)$. This likelihood is free for us to model as follows and as is standard in the *RL as inference* literature

$$p(R_k|s,a) \propto \exp \frac{Q_k(s,a)}{\eta_k}, \tag{30}$$

where $\eta_k$ is a objective-dependent temperature parameter that controls how greedy the solution $q_k(a|s)$ is with respect to its associated objective $Q_k(s,a)$. Substituting this choice of $p(R_k|s,a)$ into the independent problems (28) we obtain the following objective

$$D_{\mathrm{KL}}(q_k(a|s) \,\|\, p(R_k|s,a)\pi_i(a|s)) = \mathbb{E}_{q_k} \log \frac{q_k(a|s)}{p(R_k|s,a)\pi_i(a|s)} \tag{31}$$

$$= \mathbb{E}_{q_k} \left[ \log \frac{q_k(a|s)}{\pi_i(a|s)} - \log p(R_k|s,a) \right] \tag{32}$$

$$= D_{\mathrm{KL}}(q_k(a|s) \,\|\, \pi_i(a|s)) - \mathbb{E}_{q_k} \log p(R_k|s,a), \tag{33}$$

$$= D_{\mathrm{KL}}(q_k(a|s) \,\|\, \pi_i(a|s)) - \mathbb{E}_{q_k} \frac{Q_k(s,a)}{\eta_k}. \tag{34}$$

After multiplying this objective by $-\eta_k$, we obtain the following equivalent maximization problem

$$\max_{q_k} \mathbb{E}_\mu \mathbb{E}_{q_k} Q_k(s,a) - \eta_k \mathbb{E}_\mu D_{\mathrm{KL}}(q_k(a|s) \,\|\, \pi_i(a|s)), \tag{35}$$

where we recognize a KL regularization term that materializes naturally from this derivation. Indeed this last optimization problem can be seen as a Lagrangian, imposing a soft constraint on the KL between the improved policy $q_k$ and the current iterate $\theta_i$. If we instead impose a hard such constraint, we obtain the following problem:

$$\max_{q_k} \quad \mathbb{E}_\mu \mathbb{E}_{q_k} Q_k(s,a) \tag{36}$$

$$\text{s.t.} \quad \mathbb{E}_\mu D_{\mathrm{KL}}(q_k(a|s) \,\|\, \pi_i(a|s)) \le \epsilon_k.$$

This problem's Lagrangian form is as follows, and is nearly identical to (35)—albeit with a different dual variable instead of $\eta_k$; nevertheless we reuse the notation $\eta_k$ instead of introducing a new dual variable:

$$\mathcal{L}(q_k, \eta_k) = \mathbb{E}_\mu \mathbb{E}_{q_k} Q_k(s,a) - \eta_k \left[ \mathbb{E}_\mu D_{\mathrm{KL}}(q_k(a|s) \,\|\, \pi_i(a|s)) - \epsilon_k \right]. \tag{37}$$

**Putting everything together.** Since the original problem is equivalent to the Lagrangian relaxation of this one, we can simply substitute our improvement likelihood (30) into the solution to the original problem (29) to obtain the solution:

$$q_k(a|s) = \frac{1}{Z_k(s)} \pi_i(a|s) \exp \frac{Q_k(s,a)}{\eta_k}, \quad \text{where} \tag{38}$$

$$Z_k(s) = \int \pi_i(a|s) \exp \frac{Q_k(s,a)}{\eta_k} \,\mathrm{d}a. \tag{39}$$

**Adaptively varying $\eta_k$.** Finally, this solution can in turn be substituted back into (37) to obtain the following convex dual function:

$$\mathcal{L}^*(\eta_k) = \mathbb{E}_\mu \mathbb{E}_{q_k} Q_k(s,a) + \eta_k \mathbb{E}_\mu \left[ \epsilon_k - \mathbb{E}_{q_k} \log \frac{\frac{1}{Z_k(s)} \pi_i(a|s) \exp \frac{Q_k(s,a)}{\eta_k}}{\pi_i(a|s)} \right] \tag{40}$$

$$= \mathbb{E}_\mu \mathbb{E}_{q_k} Q_k(s,a) + \eta_k \mathbb{E}_\mu \left[ \epsilon_k - \mathbb{E}_{q_k} \log \left( \frac{1}{Z_k(s)} \exp \frac{Q_k(s,a)}{\eta_k} \right) \right] \tag{41}$$

$$= \mathbb{E}_\mu \mathbb{E}_{q_k} Q_k(s,a) + \eta_k \mathbb{E}_\mu \left[ \epsilon_k + \log Z_k(s) - \mathbb{E}_{q_k} \frac{Q_k(s,a)}{\eta_k} \right] \tag{42}$$

$$= \mathbb{E}_\mu \mathbb{E}_{q_k} Q_k(s,a) + \eta_k \mathbb{E}_\mu [\epsilon_k + \log Z_k(s)] - \mathbb{E}_\mu \mathbb{E}_{q_k} Q_k(s,a) \tag{43}$$

$$= \eta_k [\epsilon_k + \mathbb{E}_\mu \log Z_k(s)] \tag{44}$$

$$\mathcal{L}^*(\eta_k) = \eta_k \left[ \epsilon_k + \mathbb{E}_\mu \log \mathbb{E}_{\pi_i} \exp \frac{Q_k(s,a)}{\eta_k} \right], \tag{45}$$

where in the final step we substitute (39) to reveal the hidden dual variable. In practice, this function is optimized alongside of $\theta$ in order to adapt the dual variable $\eta_k$ and mitigate overfitting to $Q_k$.

### B.3 Projection or M-step

Let us now look at the projection step (M-step in EM terminology), which produces the next iterate $\theta_{i+1}$ for our parametric policy. Recall the optimization problem involved in the M-step is the following

$$\min_\theta L(\theta; q_k) = \min_\theta \sum_{k=1}^K \alpha_k \mathbb{E}_\mu D_{\mathrm{KL}}(q_k(a|s) \| \pi_\theta(a|s)), \tag{46}$$

$$= \min_\theta \sum_{k=1}^K \alpha_k \mathbb{E}_\mu \mathbb{E}_{q_k} \log \pi_\theta(a|s), \tag{47}$$

where $q_k$ is fixed to the solution from the improvement step, namely (38). This problem corresponds to finding the policy $\pi_\theta$ that acts as a barycentre between the nonparametric policies $q_k$, each weighted by its specified preference trade-off coefficient $\alpha_k$.

Other than this important difference, the multi-objective follows from the single-objective case in a straightforward way. Substituting the solution $q_k$ from (38) into the expression for $L(\theta; q_k)$ yields

$$J_{\mathrm{DiME}}(\theta) := L(\theta; q_k) = \mathbb{E}_\mu \left[ \sum_{k=1}^K \alpha_k \mathbb{E}_{q_k} \log \pi_\theta(a|s) \right] \tag{48}$$

$$= \mathbb{E}_\mu \left[ \sum_{k=1}^K \alpha_k \int \frac{1}{Z_k(s)} \pi_i(a|s) \exp \frac{Q_k(s,a)}{\eta_k} \log \pi_\theta(a|s) \, \mathrm{d}a \right] \tag{49}$$

$$= \mathbb{E}_\mu \left[ \sum_{k=1}^K \frac{\alpha_k}{Z_k(s)} \mathbb{E}_{\pi_i} \exp \frac{Q_k(s,a)}{\eta_k} \log \pi_\theta(a|s) \right] \tag{50}$$

$$= \mathbb{E}_\mu \mathbb{E}_{\pi_i} \left[ \sum_{k=1}^K \frac{\alpha_k}{Z_k(s)} \exp \frac{Q_k(s,a)}{\eta_k} \right] \log \pi_\theta(a|s). \tag{51}$$

This is the objective function that is optimized to obtain the next iterate in practice. At this stage we can further impose a trust-region or soft KL constraint on this optimization problem.

**In practice.** We compute this policy objective via Monte Carlo, sampling states $s$ from a replay buffer, then sampling actions for each state from our previous iterate $\pi_i$. We use these samples to compute weights $\exp(Q_k(s,a)/\eta_k)$ and normalize them across states, which corresponds to using a sample-based approximation to $Z_k(s)$ as well. Finally, we take the convex combination of these weights, according to the preference trade-offs, which yields the ultimate weights—seen in brackets in (51).

### B.4 ON HOW DIME IS BETTER SUITED THAN MO-MPO FOR OFFLINE RL.

In this section we expand on precisely why MO-MPO in its current published form is not readily suitable for offline RL and how our proposed method improves on it in this aspect. Let us begin by revisiting the MO-MPO objective function (Eq. 1; Abdolmaleki et al., 2020) and substituting in the offline RL objectives discussed in the paper.

$$J_{\text{MO-MPO}}(\theta) = -\mathbb{E}_\mu \left[ \sum_{k=1}^{K} D_{\text{KL}}[\pi_i e^{Q_k/\eta_k} \| \pi_\theta] \right] \tag{52}$$

$$= \mathbb{E}_\mu \left[ \sum_{k=1}^{K} \mathbb{E}_{\pi_i} e^{Q_k/\eta_k} \log \pi_\theta \right] + C \tag{53}$$

$$= \mathbb{E}_\mu \left[ \mathbb{E}_{\pi_i} e^{Q/\eta_1} \log \pi_\theta + \mathbb{E}_{\pi_i} \left( \exp \frac{\log \frac{\pi_b}{\pi_i}}{\eta_2} \right) \log \pi_\theta \right] + C. \tag{54}$$

where we deliberately omit the normalization constants to lighten the notation. Recall that MO-MPO enforces the preference trade-off across objectives by setting the KL-constraint thresholds $\epsilon_k$ between the improved policies and the current policy iterate as in (45). Therefore the preference trade-offs are implicit in the varying temperatures $\eta_k$, which are continuously adapted throughout training to ensure the satisfaction of the corresponding constraint $\epsilon_k$. As a result, when applying the same simplifying trick as used for DiME—setting $\eta_2 = 1$ and renaming $\eta_1 = \eta$—we forfeit control of the trade-off completely. Concretely, the second term above reduces to an expectation with respect to the behavior dataset $\mathcal{D}_{\pi_b}$ as follows:

$$= \mathbb{E}_\mu \mathbb{E}_{\pi_i} e^{Q/\eta} \log \pi_\theta + \mathbb{E}_{\mathcal{D}_{\pi_b}} \log \pi_\theta + C, \tag{55}$$

where we recover the now common form of an RL objective combined with an additional BC regularization term. Notice that not only is there no trade-off $\alpha$ but the temperature $\eta_2$, which implicitly controls it, has also disappeared. Let us now compare this expression to that of DiME:

$$J_{\text{DiME}} \propto \mathbb{E}_\mu \mathbb{E}_{\pi_i} e^{Q/\eta} \log \pi_\theta + \frac{\alpha}{1-\alpha} \mathbb{E}_{\mathcal{D}_{\pi_b}} \log \pi_\theta, \tag{56}$$

where we highlight the key difference between the two losses, namely that the preference parameter $\alpha$ allows us to explicitly control the relative importance of the task's Q-value objective and the objective that keeps the policy close to the one that generated the offline dataset. In contrast, MO-MPO can no longer satisfy the preference setting specified by the choice of $\epsilon_k$, because of the simplifying assumption that $\eta_2 = 1$.

An alternative approach would be to use the dataset $\mathcal{D}_{\pi_b}$ to learn a distilled behavior policy $\hat{\pi}_b$ to model $\pi_b$, effectively performing behavioral cloning as a preliminary subroutine. This would allow us to compute the log-density ratio directly, to use it as an objective in the original expression (54). This strategy introduces approximation errors in the form of assumptions made on the form of $\pi_b$ and generalization to states that are not present in $\mathcal{D}_{\pi_b}$. While we tried both this and the MO-MPO objective in (55) in preliminary experiments, neither proved to be competitive with the LS / CRR baseline and were thus omitted from further investigations.

## C  IMPLEMENTATION DETAILS

This section describes how we implemented our algorithm, DiME, and the linear scalarization (LS) and behavioral cloning (BC) baselines. To ensure a fair comparison, we use the same hyperparameters and network architectures for DiME and the baselines, wherever possible. All algorithms are implemented in Acme (Hoffman et al., 2020), an open-source framework for distributed RL.

**Training Setup.**  For the multi-objective RL and finetuning settings, we use an asynchronous actor-learner setup, with multiple actors. In this setup, actors fetch policy parameters from the learner and act in the environment, storing those transitions in the replay buffer. The learner samples batches of transitions from the replay buffer and uses these to update the policy and Q-function networks. For the offline RL setting, the dataset of transitions is given and fixed (i.e., there are no actors) and the learner samples batches of transitions from that dataset.

To stabilize learning, we use the common technique of maintaining a target network for each trained network. These target networks are used for computing gradients. Every fixed number of steps, the target network's weights are updated to match the online network's weights. For optimization, we use Adam (Kingma & Ba, 2015). For the offline RL experiments, we use Adam with weight decay to stabilize learning.

**Gathering Data.** The actors gather data and store it in the replay buffer. When the policy is conditioned on trade-offs, the trade-off is fixed for each episode. In other words, at the start of each episode, the actor samples a trade-off $\alpha'$ from the distribution $\nu$, and acts based on $\pi(a|s, \alpha')$ for the remainder of the episode. At the start of the next episode, the actor samples a different trade-off, and repeats.

**Hyperparameters and Architecture.** The policy and Q-function networks are feed-forward networks, with an ELU activation after each hidden layer. For both, after the first hidden layer, we add layer normalization followed by a hyperbolic tangent ($\tanh$); we found that this improves stability of learning. The policy outputs a Gaussian distribution with a diagonal covariance matrix.

The default hyperparameters we use in our experiments are reported in Table 3; setting-specific hyperparameters are reported in Table 4.

**Distributional Q-learning.** For our policy evaluation we use a distributional Q-network, C51 (Bellemare et al., 2017). It has proven to be a very effective critic in actor-critic methods for continuous control (Barth-Maron et al., 2018; Hoffman et al., 2020). We use the exact same training procedure as published and in open-source implementations (Barth-Maron et al., 2018; Hoffman et al., 2020), i.e., $n$-step return bootstrap targets and a projected cross-entropy loss. See Table 3 under Q-learning for our chosen hyperparameters.

**Evaluation.** We evaluate a deterministic policy, by using the mean of the output Gaussian distribution rather than sampling from it. When we evaluate a trade-off-conditioned policy, we condition it on trade-offs linearly spaced from $0.05$ to $1.0$. For each trade-off $\alpha'$, we perform several rollouts where we execute the mean action from $\pi(a|s, \alpha')$. Recall that this trade-off specifies a convex combination for two objectives.

For the multi-objective RL experiments, all policies are evaluated after 500 million actor steps. For offline RL, all policies are evaluated after 1 million learner steps.

**Learned Trade-off for Finetuning.** In our finetuning experiments, the learned trade-off is passed through the sigmoid function (i.e., $1/(1 + \exp(-x))$) to ensure it is bounded between 0 and 1.

**Compute Resources.** To train a single policy for the multi-objective RL experiments (humanoid walk and run), we used an NVIDIA v100 GPU for about 24 hours. To train a single policy for finetuning, we used a NVIDIA V100 GPU for 9-12 hours. To train a single policy for offline RL, we used a v2-32 TPU for about 6 hours for the RL Unplugged tasks, and for about 3 hours for the D4RL tasks. These computational costs were the same for all algorithms we ran: LS and the DiME variants.

# D   EXPERIMENTS: DETAILS AND ADDITIONAL RESULTS

This section provides details regarding our experiment domains, as well as additional experiments and plots.

## D.1   MULTI-OBJECTIVE RL

### D.1.1   TOY DOMAIN

Our toy domain is a bandit with a continuous action $a \in \mathbb{R}$. The reward $r(a) \in \mathbb{R}^2$ is specified by either the Schaffer function or the Fonseca-Fleming function, $f : \mathbb{R} \mapsto \mathbb{R}^2$. Each point along this function is optimal for a particular tradeoff between the two reward objectives, so the function can be seen as a Pareto front.

The Schaffer function corresponds to a convex Pareto front, and is defined as

$$f_1(a) = a^2 \quad \text{and} \quad f_2(a) = (a - 2)^2 .$$

| Category | Hyperparameter | Default |
|---|---|---|
| training setup | batch size | 512 |
| | number of actors | 4 |
| | replay buffer size | $3 \times 10^{-7}$ |
| | target network update period | 100 |
| | Adam learning rate | $10^{-4}$ |
| policy & Q-function networks | layer sizes | $(1024, 1024, 1024, 1024, 1024, 512)$ |
| Q-learning | support | $[-150, 150]$ |
| | number of atoms | 101 |
| | n-step returns | 5 |
| | discount $\gamma$ | 0.99 |
| policy loss | actions sampled per state | 30 |
| | KL-constraint on $q_k(a|s)$, $\epsilon_k$ | 0.1 |
| | KL-constraint on policy mean, $\beta_\mu$ | 0.0025 |
| | KL-constraint on policy covariance, $\beta_\Sigma$ | $10^{-5}$ |
| | initial temperature $\eta$ | 10 |
| | Adam learning rate (for dual variables) | $10^{-2}$ |

Table 3: Default hyperparameters for all approaches, with decoupled KL-constraint on mean and covariance of the policy M-step.

| **Multi-Objective RL** | | |
|---|---|---|
| training setup | number of actors | 32 |
| | replay buffer size | $10^6$ |
| | target network update period | 200 |
| experiment setup | trade-offs (DiME) | $\alpha_{\text{task}} = 1$ |
| | | $\alpha_{\text{penalty}} \in \text{linspace}(0.3, 1.5)$ |
| | trade-offs (MO-MPO) | $\epsilon_{\text{task}} = 0.1$ |
| | humanoid walk | $\epsilon_{\text{penalty}} \in \text{linspace}(0, 0.3)$ |
| | humanoid run | $\epsilon_{\text{penalty}} \in \text{linspace}(0, 0.15)$ |
| | trade-offs (LS) | $\alpha_{\text{task}} = 1 - \alpha_{\text{penalty}}$ |
| | | $\alpha_{\text{penalty}} \in \text{linspace}(0, 0.15)$ |
| policy loss | actions sampled per state | 20 |
| | KL-constraint on policy mean, $\beta_\mu$ | 0.001 |
| | KL-constraint on policy covariance, $\beta_\Sigma$ | $10^{-7}$ |

| **Offline RL** | | |
|---|---|---|
| training setup | Adam weight decay (RL Unplugged) | 0.999999 |
| | Adam weight decay (D4RL) | 0.99999 |
| experiment setup | trade-offs | $\alpha \in \text{linspace}(0.005, 1)$ |
| | trade-off distribution, $\nu$ | $\mathcal{U}(0.005, 1)$ |
| policy loss | KL-constraint $\epsilon$ (LS) | 0.01 |
| | KL-constraint $\epsilon$ (RL Unplugged) | 0.5 |
| | KL-constraint $\epsilon$ (Kitchen) | 0.001 |
| | KL-constraint $\epsilon$ (Maze, DiME multi) | 0.5 (BC), 0.01 (AWBC) |

| **Finetuning** | | |
|---|---|---|
| training setup | batch size | 1024 |
| experiment setup | initial trade-off (for learned trade-offs) | 0.5 |

Table 4: Hyperparameters that are either setting-specific or differ from the defaults in Table 3.

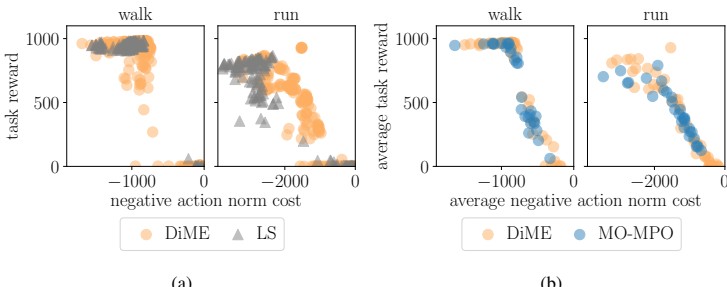

(a)                      (b)

Figure 4: (a) This plot shows the per-episode task reward and action norm cost, rather than the average across episodes. Note that the linear scalarization (LS) cannot find solutions along the Pareto front for walk, because it is likely concave. In contrast, DiME can find solutions that compromise between the two objectives. (b) DiME finds similar Pareto fronts as MO-MPO, a state-of-the-art multi-objective RL algorithm.

The Fonseca-Fleming function corresponds to a concave Pareto front, and is defined as

$$f_1(a) = 1 - \exp(-(a-1)^2) \quad \text{and} \quad f_2(a) = 1 - \exp(-(a+1)^2).$$

These are standard test functions in multi-objective optimization, where the aim is to minimize them (Okabe et al., 2004). Since in RL we want to maximize reward, we define the reward to be the negative of these functions.

### D.1.2 HUMANOID

We also evaluate on the humanoid walk and run tasks from the open-sourced DeepMind Control Suite (Tassa et al., 2018), with the two objectives proposed in Abdolmaleki et al. (2020). The first objective is the original task reward, which is a shaped reward that is given for moving at a target speed (1 meters per second for walking and 10 m/s for running), in any direction. The second objective is the negative $\ell 2$-norm of the actions, $-\|a\|_2$, which can be seen as encouraging the agent to be more energy-efficient.

The observations are 67-dimensional, consisting of joint angles, joint velocities, center-of-mass velocity, head height, torso orientation, and hand and feet positions. The actions are 21-dimensional and in $[-1, 1]$; these correspond to setting joint accelerations. Each episode has 1000 timesteps. At the start of every episode, the configuration of the humanoid is randomly initialized.

**Additional Results.** When we plot the per-episode performance (rather than an average across episodes), we see that linear scalarization only finds solutions at the two extremes for the humanoid walk task, i.e. policies that achieve high task reward but with high action norm cost, or incur zero cost but fail to obtain any reward (Fig. 4a). We hypothesize that this may be because the humanoid walk task has a concave Pareto front of solutions; recall that linear scalarization is fundamentally unable to find solutions on a concave Pareto front (Das & Dennis, 1997). In contrast, DiME is able to find solutions that trade off between the two objectives.

On both tasks, DiME finds a similar Pareto front as MO-MPO, which is a state-of-the-art approach for multi-objective RL (Fig. 4b).

DiME requires training a separate policy per trade-off, which can be computationally expensive. In contrast, our trade-off-conditioned version of DiME, which we call DiME multi, trains a *single* policy for a range of trade-offs. This obtains a similar Pareto front compared to that found by DiME (Fig. 5), after the same number of actor steps (500 million).

We also ran an experiment to test the scale-invariance of DiME. We modified the humanoid run task such that the action norm cost is now multiplied by ten. We used DiME and linear scalarization to train policies for three random seeds, for each of two trade-off settings. For both the original task and the task with scaled-up action norm costs, DiME trains policies with similar performance (Fig. 6, right). This is the case for both trade-off settings—note the two clusters of differently-colored points in the plot. In contrast, linear scalarization is not scale-invariant: task performance suffers when the cost is scaled up (Fig. 6, left). With a trade-off of 0.05, although this obtains better performance than

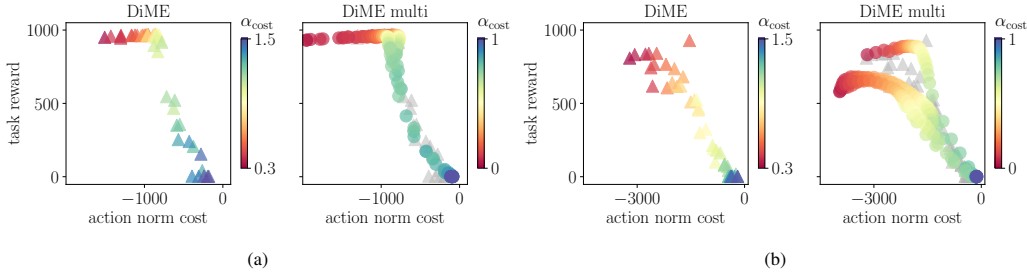

(a)                                                              (b)

Figure 5: DiME multi trains a single trade-off-conditioned policy, that finds a similar Pareto front as DiME does for humanoid (a) walk and (b) run. Each plot for DiME multi shows five trained policies, for different random seeds; these seeds vary somewhat in their performance because the policy learns different strategies (e.g., spinning, running sideways, etc.). Each policy is conditioned on trade-offs $\alpha$ linearly spaced between 0 and 1. The color denotes the trade-off. For DiME, $\alpha_{\text{task}} = 1$; DiME multi uses a convex combination (i.e., $\alpha_{\text{task}} = 1 - \alpha_{\text{cost}}$).

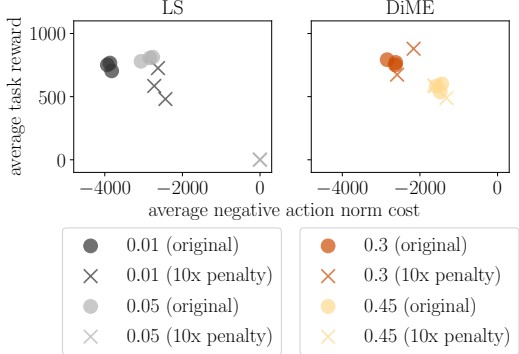

Figure 6: With the original objective scales for humanoid run, both linear scalarization (left plot) and DiME (right) obtain similar performance. When the action norm cost is scaled up by ten times, DiME's performance is unaffected because its trade-off setting is invariant to reward scales—note the two clusters of differently-colored points. The color denotes the tradeoff $\alpha$ and the marker symbol denotes whether the action norm cost is scaled by ten times. Note that in the plots, the y-axis is the unscaled value of action norm cost.

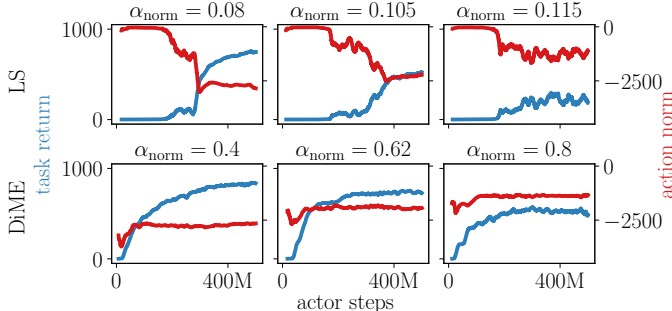

Figure 7: Per-objective learning curves for humanoid run. During training, DiME (below) is able to improve on both objectives simultaneously. In contrast, LS (above) can typically only improve on one objective at a time. This helps DiME find better solutions—in each column, the LS and DiME policies converge to similar action norm cost, but the DiME policy performs substantially better on the task.

the trade-off of 0.01 with the original objectives (i.e., gets higher reward with lower cost), with the scaled-up action norm cost, policies do not obtain any task reward at all with a trade-off of 0.05.

Finally, we delve into *why* DiME outperforms LS. Our hypothesis is that this is because DiME computes an improved policy for each objective (before projecting them), and is thus more likely to ensure improvement with respect to all objectives. In contrast, since LS combines rewards directly, it cannot guarantee improvement with respect to all objectives. To test out this theory, we plotted how policies trained by LS and DiME perform with respect to each objective over the course of training, for humanoid run. We found that policies trained with DiME indeed exhibit improvement

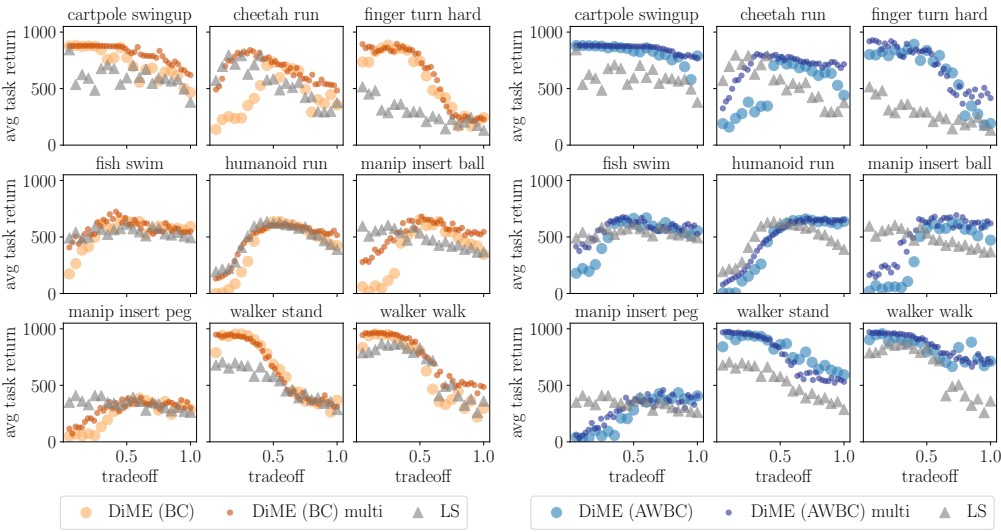

Figure 8: Per-tradeoff performance for all nine Control Suite tasks from RL Unplugged. Across all nine tasks, the best solution found by DiME (any variation) obtains either higher or on-par task performance than the best found by linear scalarization (LS). In addition, DiME can train a *single* policy for a range of tradeoffs (DiME multi, dark orange and dark blue), that performs comparably to learning a separate policy for each tradeoff (orange and blue), after the same number of learning steps.

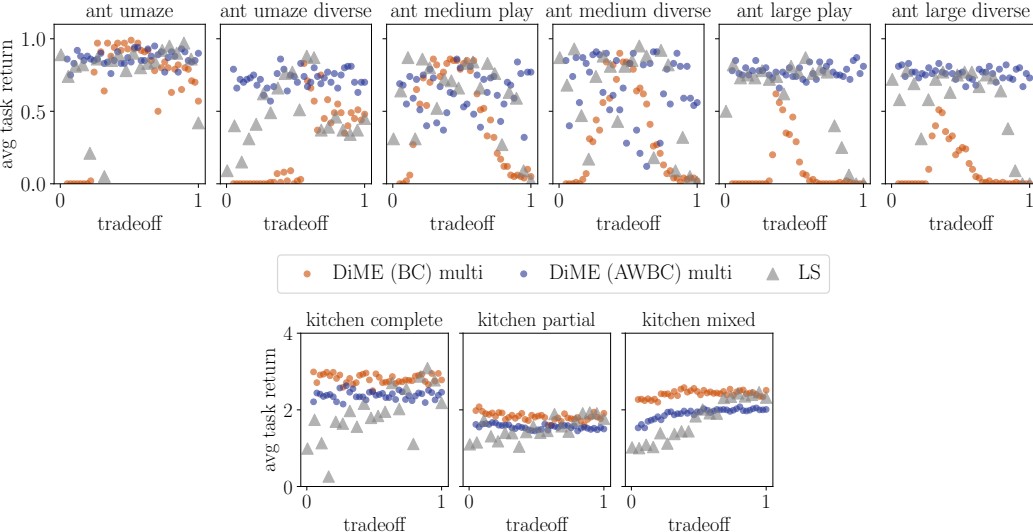

Figure 9: Across nine tasks from D4RL, DiME and LS (CRR) perform on-par.

with respect to both objectives simultaneously over the course of training, whereas policies trained with LS typically suffer from worsening performance for one objective while improving for the other.

## D.2 OFFLINE RL

In our offline RL experiments, we evaluate on two open-sourced benchmarks for offline RL: RL Unplugged (Gulcehre et al., 2020) and D4RL (Fu et al., 2020). We use the nine Control Suite tasks from RL Unplugged, the six Ant Maze tasks from D4RL, and the three Franka Kitchen tasks from D4RL. For each task, we train policies on the dataset of transitions provided by the benchmark. These are continuous control tasks, that include particularly challenging ones that state-of-the-art approaches perform poorly on (e.g., Ant Maze large play and large diverse). For a comprehensive description of the environments, please refer to Gulcehre et al. (2020) and Fu et al. (2020).

| Task | *BRAC* | *BEAR* | *AWR* | *BCQ* | *CQL* | BC | LS (CRR) $\alpha$=.63 | LS (CRR) | DiME (BC) multi | DiME (AWBC) multi |
|------|--------|--------|-------|-------|-------|------|------|------|------|------|
| antmaze umaze | 0.7 | 0.7 | 0.6 | 0.8 | 0.7 | 0.42 | 0.83 | **0.97** | **0.99** | **0.95** |
| antmaze umaze diverse | 0.7 | 0.6 | 0.7 | 0.6 | 0.8 | 0.45 | **0.87** | **0.88** | 0.83 | 0.86 |
| antmaze medium play | 0 | 0 | 0 | 0 | 0.6 | 0.02 | 0.66 | **0.87** | **0.86** | **0.86** |
| antmaze medium diverse | 0 | 0.1 | 0 | 0 | 0.5 | 0.01 | 0.88 | **0.95** | **0.90** | **0.91** |
| antmaze large play | 0 | 0 | 0 | 0.1 | 0.2 | 0 | 0.76 | **0.87** | 0.62 | **0.87** |
| antmaze large diverse | 0 | 0 | 0 | 0 | 0.1 | 0 | 0.67 | **0.78** | 0.51 | **0.83** |
| kitchen complete | 0 | 0 | 0 | 0.3 | 1.8 | 2.18 | 2.70 | **3.09** | **2.99** | 2.63 |
| kitchen partial | 0 | 0.5 | 0.6 | 0.8 | 1.9 | 1.77 | 1.89 | **1.93** | **2.08** | 1.67 |
| kitchen mixed | 0 | 1.9 | 0.4 | 0.3 | 2.0 | 2.32 | 1.90 | **2.46** | **2.58** | 2.08 |
| mean score | 0.16 | 0.42 | 0.26 | 0.32 | 0.96 | 0.80 | 1.24 | **1.42** | **1.37** | 1.30 |

Table 5: Results for LS and DiME (average cumulative reward) obtained for the best tradeoff per algorithm, on tasks from D4RL. The results for the italicized algorithms are taken from Fu et al. (2020); these are the best-performing algorithms from their comprehensive evaluation. Numbers within 5% of the best score are in bold.

**Additional Results.** Here we show plots for per-tradeoff performance for all nine Control Suite tasks from RL Unplugged (Fig. 8) and for the Ant Maze and Franka Kitchen tasks from D4RL (Fig. 9). We also provide the full table of results for the D4RL tasks (Table 5). Across all 18 tasks, the best tradeoff found by DiME obtains either higher or on-par performance compared to the best tradeoff found by LS. Recall that LS is equivalent to CRR, a state-of-the-art approach for offline RL, in terms of the policy loss it optimizes for.

## D.3 FINETUNING

For finetuning, we evaluate on three humanoid tasks (stand, walk, and run) and two manipulator tasks (insert ball and insert peg) from the DeepMind Control Suite. There are two main differences between finetuning and offline RL: 1) in finetuning the policy is able to interact with the environment, whereas in offline RL it cannot, and 2) in finetuning the behavioral prior may be in the form of a teacher policy that we can query to obtain $\pi_b(a|s)$, whereas in offline RL the behavior prior is in the form of a dataset of transitions.

For the humanoid tasks, we trained a policy for humanoid stand with standard deep RL (using MPO (Abdolmaleki et al., 2018)) and stopped it midway through training, when it reached an average task reward of about 400. We then used this teacher policy for finetuning in all three humanoid tasks. However, for the manipulator tasks, training a policy from scratch is highly sample-inefficient, as can be see by the learning curves for learning from scratch (i.e., $\alpha = 0$) in Fig. 10. So for manipulator insert ball and insert peg, we took the best policy trained using DiME in our offline RL experiments, and used that as the teacher policy for finetuning in that task.

Fig. 10 shows learning curves for all five tasks. When the tradeoff is learned (in orange), DiME and linear scalarization perform on-par. However, when the tradeoff is fixed, for linear scalarization it is more difficult to pick an appropriate tradeoff, because the appropriate tradeoff depends on the relative scales of the rewards. In fact, for the humanoid tasks, none of the fixed (non-zero) tradeoffs for LS perform better than learning from scratch.

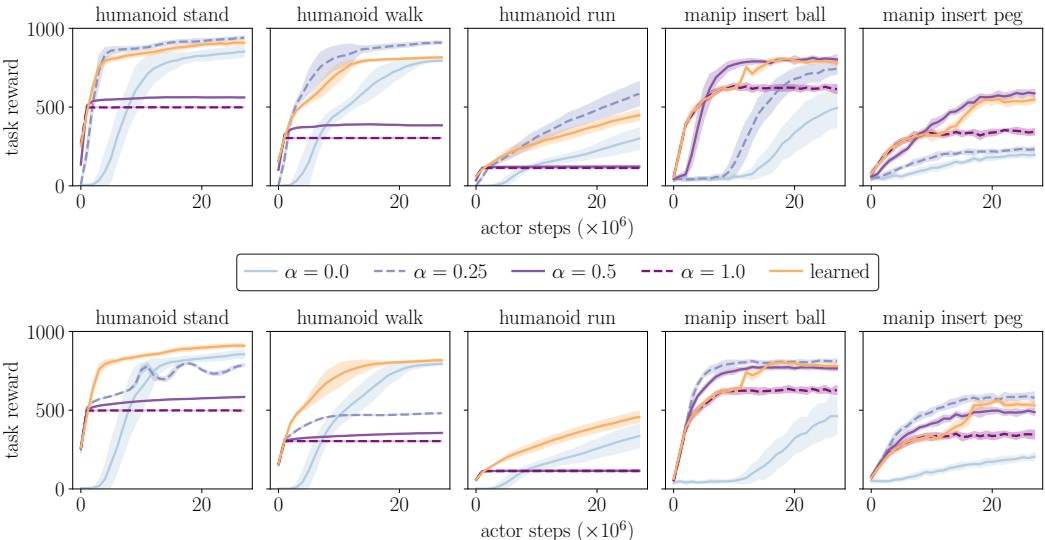

Figure 10: Learning curves for finetuning, for DiME (top row) and linear scalarization (LS, bottom row). $\alpha = 1$ corresponds to fully imitating the behavioral prior, while $\alpha = 0$ corresponds to learning from scratch. The optimal fixed tradeoff $\alpha$ depends on the task. For both DiME and LS, learning the tradeoff (orange) converges to better performance than fully imitating the behavioral prior (dashed purple), while learning as quickly. The error bars show standard deviation across ten seeds.

