# OpenReview forum: "On Multi-objective Policy Optimization as a Tool for Reinforcement Learning: Case Studies in Offline RL and Finetuning"
_ICLR.cc/2022/Conference — ICLR 2022 Submitted_

### Official Review · Reviewer_JB6Q · 2021-11-02

**Correctness:** 4
**Technical Novelty And Significance:** 2
**Empirical Novelty And Significance:** 2
**Recommendation:** 6
**Confidence:** 3

**Main Review:**

Strength

- The idea can be naturally applied to the offline rl and finetuning setting, since in these two problems, we can view the original objectives as two different objectives when learning the policy.
- The experiments are conducted thoroughly and demonstrate the effectiveness of the current approach.

Weakness

- The computational costs. Since the new proposed method requires to sample alpha to learn, it may introduce additional computational costs to learn the multiobjective, especially when we need to learn the whole pareto. Would the authors discuss the computational costs than the single linear combination baseline methods?

- I wonder if it is possible for the authors to compare some other multi objective baselines instead of linear scalarization, such as multiple gradient descent or pcgrad? since I think linear scalarization is the most native baseline for multi objective optimization.

- I am always curious how the authors implemented the multiobjective optimization part, ever since your first paper. But I never see an opensource code and it is really hard to get the intuition and computation efficiency of the proposed method just from what you described in the appendix.



**Summary Of The Paper:**

The authors in this paper propose to utilize multi-objective optimization techniques to improve offline reinforcement learning and finetunning.
Based on the extensive empirical experiments, the proposed method can outperform other baselines which do not view the problem as a multi-objective optimization problem.


**Summary Of The Review:**

Overall I think it is interesting to introduce multiobjective to improve offline rl and policy finetuning tasks, although the computational costs might increase a lot compare with the native approach.

---

> ### Author Response · Authors · 2021-11-17
> **Author Response**
>
> **Computational costs**
>
> We agree that computational cost is very important to consider. In fact, the computational costs for LS, DiME, and DiME multi (i.e., $\alpha$-conditioned) are all very similar.
>
> It might seem surprising that learning the entire Pareto front is not more computationally expensive than finding a single solution on the Pareto front. This is because for each policy improvement step, DiME multi just augments each sampled state with a single different sampled $\alpha$, before passing it through the Q-function and policy networks, so the batch size stays the same. (This is described in Section 4 and equation (4).) In practice, learning the entire Pareto front can lead to more diverse exploration and can also help regularize learning of the Q-function and policy, both of which can benefit learning.
>
> We took computational cost into account to make sure we compared algorithms fairly:
> * In our offline RL experiments, all LS, DiME, and DiME multi policies are evaluated after the same number of learner steps. In our MORL experiments, we evaluated all algorithms after the same number of actor steps. In our finetuning experiments, the learning curves show the sample efficiency with respect to actor steps required.
> * All algorithms share the same learning hyperparameters (e.g., batch size) and network sizes.
> * We found that the training walltime is quite similar for all algorithms. We report the walltimes and compute resources used at the end of Appendix C.
>
> **Additional MORL baselines**
>
> We agree that LS is the simplest MORL algorithm, and there exist many MORL algorithms that outperform it. Please take a look at the main rebuttal, where we explain why we did not compare to MORL algorithms that cannot be applied to offline RL, which includes gradient-based approaches like multiple-gradient descent and PCGrad. While we agree that including more baselines and benchmarks in our MORL experiments would be useful, for this paper we instead prioritized running extensive experiments to evaluate our key claim, by showing the effectiveness of applying DiME for offline RL and finetuning.
>
> **Implementation of multi-objective component**
>
> The authors of MO-MPO open-sourced that algorithm in the Acme framework [1]. We built on this open-source implementation for DiME. We are working on open-sourcing DiME, as well as the novel algorithms for offline RL and finetuning that we proposed.
>
> **Final thoughts**
>
> Please let us know if this response addresses your concerns, and if not, what we can do to clarify any remaining questions.
>
> ------------
>
> [1] https://github.com/deepmind/acme/tree/master/acme/agents/tf/mompo

---

### Official Review · Reviewer_gxGj · 2021-11-02

**Correctness:** 3
**Technical Novelty And Significance:** 2
**Empirical Novelty And Significance:** 2
**Recommendation:** 3
**Confidence:** 3

**Main Review:**

**Strengths**

The experimental study in this paper is quite extensive, with results on two important applications (offline RL, finetuning) and various task domains. From Table 1 and Table 2, it is quite convincing that DiME outperforms the baseline method of LS (and other existing non multi-objective based algorithms) on offline RL.

The presentation of this paper is quite clear.

**Weaknesses**

My central concern about this paper is about how DiME compares with the very related existing method of MO-MPO. I have two main questions on this end.
1. Experimentally, MO-MPO is not tested in the main experimental results (Table 2, Figures 2-3). Given that MO-MPO is presented as one of the two main existing algorithms in Table 1 (alongside LS), and LS is tested quite extensively in the above tables/figures, it is somewhat surprising that MO-MPO is not tested in the above results. Only Section 6.1 (multi-objective RL) mentions MO-MPO with results in appendix D, showing that DiME only performs on par with MO-MPO.
Section 5.2 contains a short discussion on why MO-MPO is not suitable for offline RL. However I find the explanations quite vague and not convincing why it is not implementable or necessarily performs badly in offline RL.

2. Algorithmically, I feel like DiME may actually be not different at all from MO-MPO, especially after taking into account an important implementation trick hidden in the appendix.
Comparing DiME vs. MO-MPO in Table 1 and Section 3.1, DiME involves choosing both the stepsize $\eta_k$ as well as the weights $\alpha_k$ for the $K$ objectives, whereas for MO-MPO the step-size $\eta_k$ is determined from $\alpha_k$. However, when taking into account the actual implementation details, $\eta_k$ in DiME are actually optimized too, in a fashion very similar to MO-MPO indeed.
For example, the DiME objective (11) for offline RL has $K=2$, and it chooses $\eta_2=1$ but still leaves $\eta_1=\eta$ a hyperparameter. The main paper and the implementation detail section in Appendix C only provides the initial value of $\eta$, but does not further tell how it is chosen or tuned.
Going through the appendix, the way to choose $\eta_k$ seems to be hidden in Section B.1, Page 17 (which ought to be a section for the math intuition about variational inference, not for implementation details). From there it seems like $\eta_k$ needs to be optimized with gradient-based optimization on the dual objective (45)---This makes it exactly the same to Eq (4) of the MO-MPO paper (Abdolmaleki et al. 2020). Thus I am afraid that DiME is in essence almost the same as MO-MPO.

Regarding these two questions, I’m curious whether the authors could either provide some more details on how the actual implementation of DiME differs from MO-MPO, and/or present some experiments on the MO-MPO algorithm in conjunction with DiME and LS.


**Summary Of The Paper:**

This paper proposes a new algorithm DiME for multi-objective policy optimization. The main target application is in the problems of offline RL and policy finetuning, where some closeness to the behavior policy is often desired, in conjunction with optimizing the value function. Experimentally, the paper shows that DiME outperforms the baseline method of Linear Scalarization (LS) on the above applications, and presents theoretical insights on the comparisons between the proposed DiME and several existing algorithms.

**Summary Of The Review:**

Overall, this is a well-executed paper that presents a new algorithm DiME for multi-objective policy optimization. However, currently, I am afraid it is missing comparisons with the very related prior method MO-MPO, and hiding important implementation details with which the present algorithm becomes almost equivalent to that prior method.

---

> ### Author Response · Authors · 2021-11-17
> **Author Response**
>
> **Why MO-MPO cannot be applied to offline RL**
>
> We have now added a section (Appendix B.4) that explains precisely why MO-MPO cannot be applied to offline RL, with step-by-step derivations. In a nutshell, it is because in offline RL we cannot compute the log-density ratio for the objective of staying close to the dataset. There are two ways to get around this while still using MO-MPO, which we describe in Appendix B.4, but neither is desirable. We actually started off by trying both of these, but the first was not competitive with the LS / CRR baseline, and the second (that involves behavioral cloning) was very complicated to implement and get working. We realized that a small but crucial change to MO-MPO made it easier to apply to offline RL, and this is how we came up with DiME.
>
> **DiME is very similar to MO-MPO**
>
> DiME is indeed quite similar to MO-MPO. This is why we included Table 1, to highlight the key difference. To clarify, optimizing the temperature $\eta_k$ on the dual objective (45) is the standard technique for computing the temperature for the closed-form solution of $q_k(a|s)$ given in Table 1. LS also uses this to compute the temperature. In the literature this technique is used by not only MO-MPO, but also REPS, MPO, and VMPO, to give a few examples. We certainly did not intend to hide how $\eta_k$ is computed, and have updated the paper to make it more clear where this is explained. The reason we put this in the Appendix is because it is not an important component of DiME, since DiME does not rely on adapting $\eta_k$ to specify the trade-off across objectives. In contrast, MO-MPO *does* rely on adapting $\eta_k$ to specify the trade-off.
>
> More concretely, for objective $k$, $\eta_k$ is optimized in order to ensure that the KL-divergence between the improved policy $q_k$ and the previous policy iterate $\pi_i$ is less than $\epsilon_k$, as shown in equations (36) and (37) in the Appendix. In MO-MPO, we set the $\epsilon_k$ in order to specify a trade-off across objectives, which is analagous to setting reward weights in LS and setting $\alpha_k$ in DiME. Thus, MO-MPO must be able to adaptively vary the temperatures $\eta_k$ in order to satisfy the trade-off over objectives that $\epsilon_k$ defines. In contrast, for DiME we can set both the $\epsilon_k$ and the $\alpha_k$. However, in our experiments, we keep $\epsilon_k$ fixed and use solely the $\alpha_k$ to specify the trade-offs. ***Thus, DiME does not rely on varying the temperatures $\eta_k$ to specify the trade-off across objectives, and this is why it can be applied to offline RL whereas MO-MPO cannot.***
>
> We called this algorithm Distillation of a Mixture of Experts (DiME) because it is an intuitive name, but we would be happy to change the name to be MO-MPO++ to make the connection more obvious.
>
> **Final thoughts**
>
> Please let us know if this response and the updated paper address your concerns, and if not, what we can do to clarify any remaining questions.

---

### Official Review · Reviewer_gFyL · 2021-11-08

**Correctness:** 4
**Technical Novelty And Significance:** 3
**Empirical Novelty And Significance:** 3
**Recommendation:** 6
**Confidence:** 3

**Main Review:**

The paper studies a foundamental problem in RL optimization with an interesting view of multi-objective policy optimization, trying to tackle challenges in existing RL approaches.
The main contributions of this work are in two-fold. First, it provides a novel view that uses MORL as a tool to address fundamental challenges in RL, and in the paper, as case studies, the authors studies the use of MORL in finetuning and offline RL. Second, it proposes a novel MORL approach DiME that outperforms tranditional LS on standard MORL tasks.

The paper did a good derivation and analysis for the proposed DiME approach. Table 1 is nice to show the comparison between prior works. However, it is not theoretically shown that why DiME can outperform LS, though in the paper the authors hypothesize that a separate variational distribution allows DIME to find a tighter variational lower bound. In the experiments, the paper shows that DiME outperforms LS on standard MORL tasks empirically. However, it did not show comparison to MO-MPO --- this is wierd because MO-MPO seems to serve as a better baseline than LS.

In section 5, the paper did case studies of MORL for finetuning and offline RL, and showed how to formulate the algorithms of finetuning and offline RL as MORL problem. The authors provide a general formulation and shows that small modifications to the general objective can recover to existing algorithms. Extensive experiments show that DiME allows better optimization for finetuning algorithms and offline RL algorithms.


**Summary Of The Paper:**

This paper proposes using multi-objective optimization as a tool for tackling challenges in RL. The motivation is that the different additional contraints (or objectives) in the policy optimization step are always in conflict, which is natural to view the existing RL as an instance of MORL problem. Typical MORL applies linear scalarization (LS) by taking a weight sum of the objectives. In this paper, first, the authors identifies the disadvantages of prior work (including LS and MO-MPO) and proposes a new objective called DiME. The proposed DiME fits a separate variational distribution per objective and thus can find a tighter variational lower bound. Second, the authors show case studies of DiME for finetuning and offline RL. Experiments show that DiME outperforms LS on standard MORL problems as well as two case studies (finetuning and offline RL).


**Summary Of The Review:**

Pros:
1. Provide an interesting perspective that regards RL as MORL problems.
2. Propose a better MORL algorithm DiME.
3. Experimental results are interesting and convincing.

Cons:
1. Lack of theoretical results.
2. Did not compare with all baselines, such as MO-MPO.

---

> ### Author Response · Authors · 2021-11-17
> **Author Response**
>
> **Missing comparison to MO-MPO**
>
> The Appendix contains a comparison of DiME against MO-MPO, on the challenging humanoid tasks (Appendix D.1.2, Figure 4b). Here DiME performs on-par with MO-MPO. We left this comparison for the Appendix, since it is tangential to the key claim of our paper. Please take a look at our main rebuttal, which clarifies the key claim of our paper.
>
> **Theoretical basis for why DiME outperforms LS**
>
> We agree that it would be valuable to have a theoretical understanding of why DiME outperforms LS, beyond the RL-as-inference derivations. We are very interested in working on this direction in the future, but it is outside the scope of this work. In this work, we focus on: 1) drawing connections between DiME and the well-established LS approach; 2) applying LS and DiME to offline RL and finetuning, which leads both new and existing policy update rules; and 3) showing strong empirical results for our new update rules.
>
> **Final thoughts**
>
> Please let us know if this response addresses your concerns, and if not, what we can do to clarify any remaining questions.

---

### Official Review · Reviewer_kKtt · 2021-11-09

**Correctness:** 3
**Technical Novelty And Significance:** 2
**Empirical Novelty And Significance:** 3
**Recommendation:** 5
**Confidence:** 3

**Main Review:**

1. Between weighing the Q-values directly in the policy update vs. weighing KL distances, as DiME does, the paper repeatedly points to the fact that the former is sensitive to the magnitude of rewards. Is there a more fundamental distinction in the tasks one can perform better on vs. the other, especially for a fixed scalar reward (the abstract seems to make a case for this; but the justification for this is not followed up on the paper)? Possibly in terms of some qualitative or quantitative features one might ascribe to the task?
2. The fact that the proposed approach can discover the pareto frontier  for parameterized rewards is impressive. But, as I understand, both LS (which may have similarities to existing approaches) & DiME are both algorithms proposed in this paper. Is a more direct comparsion possible to an exisiting, published result? MORL is a quite a well established problem in RL. Is there a reason why considering MO-MPO's (for instance) experimental setup beyond humanoid walk-run (for which results are fairly similar) for MORL are prohibitive here?

**Summary Of The Paper:**

Modern day policy (and value) function optimization procedures routinely include terms beyond the typical scalar reward, such as to induce regularization, encourage exploration and perform discriminative feature learning. How must one go about balancing these objectives? The typical scheme is to take a weighted sum; this paper recasts that into a multi-objective optimization question, and proposes an objective that combines policies from various objectives in a weighted KL sense.

**Summary Of The Review:**

Futher comparisons on existing benchmarks along with characterization (empirically is more than good enough) of environment where the proposal works would really add value.

---

> ### Author Response · Authors · 2021-11-17
> **Author Response**
>
> **Additional MORL baselines and benchmarks**
>
> We ask the reviewer to please take a look at the main rebuttal, where we explain why comparing to more MORL algorithms and on more MORL tasks is tangential to the key claim of this paper. While we agree that including more baselines and benchmarks in our MORL experiments would be useful, for this paper we instead prioritized running extensive experiments to evaluate our key claim, by showing the effectiveness of applying DiME for offline RL and finetuning.
>
> **Where DiME outperforms LS**
>
> There are three main situations where DiME outperforms LS, as listed below. For the second point, we added analysis plots as requested by the reviewer, in Appendix D.1.2 (Figure 7).
> 1. **Concave:** LS is unable to find solutions on the concave portions of a Pareto front, whereas DiME can. Prior work proved that this is a fundamental limitation of LS (Das & Dennis 1997). Empirically, we showed DiME can find solutions on concave Pareto fronts, whereas LS cannot, for both the Fonseca-Fleming toy task and humanoid walk (Figure 1).
> 2. **Strongly conflicting objectives:** DiME outperforms LS on tasks with strongly conflicting objectives. The humanoid run task has strongly conflicting objectives, because minimizing energy usage makes it much harder for the agent to run quickly. Empirically, for this task we can see that DiME is better than LS at finding solutions that do well at both objectives; these solutions are those in the upper right corner of the Pareto plot (Figure 1b). Intuitively, this is because DiME improves the policy separately with respect to each objective. In contrast, LS does not guarantee that the policy improves with respect to every objective, and in practice, typically improving performance on one objective comes at the cost of decreasing performance on another. As requested, we have added empirical support for this point, by plotting the learning curve for each objective during training (end of Appendix D.1.2, Figure 7).
> 4. **Scale-invariance:** DiME is scale-invariant, whereas LS is not. In other words, when the scales of the rewards change, the same LS weights now correspond to a different trade-off across objectives. This is because LS directly combines the rewards, whereas DiME combines distributions which are normalized. Being dependent on reward scales can be problematic. For example, typically the scale of the obtained task reward will change over the course of training, as the agent learns to solve the task. This means that the same LS weights will define a different trade-off across objectives at the beginning of training, compared to the middle or end. Empirically, we show that preference settings in DiME correspond to similar trade-offs, even when one of the objectives' rewards is scaled up by ten times. This is not the case for LS (Figure 6 in Appendix).
>
> **Final thoughts**
>
> Please let us know if this response and the updated paper address your concerns, and if not, what we can do to clarify any remaining questions.

---

### Author Response · Authors · 2021-11-17
**Main rebuttal**

We thank the reviewers for their time and valuable comments. We are glad the reviewers agree that the experiments are extensive and show the value of using a multi-objective approach for offline RL and policy finetuning.

However, we were suprised that most reviewers focused on the multi-objective RL (MORL) aspect of this work, and how that could be improved. This is tangential to the point of our paper - this is not a typical MORL paper, that introduces a new MORL algorithm and tries to show it is the new state-of-the-art on MORL tasks.

Instead, this is a paper that introduces a new perspective that gives rise to novel update rules. ***Our key claim is that it is valuable to apply MORL to those general problems in RL that can be seen as optimizing for different objectives simultaneously.*** We designed our experiments with this claim in mind. Through this lens, we proposed novel update rules for offline RL and finetuning, that are principled and outperform state-of-the-art approaches. For perspective, recent publications in offline RL propose a single novel update rule, e.g. CRR and TD3-BC both at NeurIPS.

***We hope that the reviewers can reconsider their view of this paper as a typical MORL paper, and instead evaluate it based on whether they believe our key claim is useful and supported.***

**Choice of MORL baselines**

All reviewers asked why we did not compare to other MORL algorithms on the MORL, offline RL, and finetuning tasks. This is a valid question, since there are indeed many existing MORL algorithms that outperform linear scalarization (LS). However, most of these algorithms require access to the rewards/Q-values. Offline RL does not satisfy this requirement, so most existing MORL algorithms cannot be applied, including MO-MPO and gradient-based approaches.$^1$ We have updated Section 5.2 to clarify this.

It is also worth noting that although LS is a naive solution for MORL tasks, applying LS to offline RL and finetuning is not naive. In fact, recently-proposed offline RL algorithms (e.g., CRR and AWAC) are essentially applying LS to offline RL, as we showed in Section 5.2.

In the MORL experiments, we did not think it was worthwhile to compare DiME against MORL algorithms that could not be applied to the offline RL setting. (Although we do compare to MO-MPO, in Appendix D.1.2.) This is because our aim is not to show that DiME is a state-of-the-art MORL algorithm. Instead, we aim to demonstrate DiME's value in being applied to offline RL and finetuning, in order to support our key claim. We have updated Section 6 to clarify this.

**Development of DiME**

For context, when we started thinking about using MORL to tackle offline RL, we first tried applying existing MORL algorithms. We initially tried using MO-MPO, but it was very complicated to apply this to offline RL (we explain why in Appendix B.4). We realized that a small but crucial change to MO-MPO resulted in a much simpler algorithm. This is what led us to DiME. One can think of DiME as MO-MPO++, with the additional ability to set the trade-offs by choosing the weights $\alpha_k$ in the projection step (Table 1). Since this small change led us to the unification of LS and MO-MPO under a single MORL framework (i.e., as simply two different choices of variational factorizations), we opted for a new overarching and descriptive name.

**Additional questions**

We will address other questions and concerns in individual replies to reviewers. We have also uploaded a new version of the paper, with changes in blue, based on the reviewers' input.

--------

$^1$ For offline RL, the reward for the objective of staying close to the dataset (i.e., the log-density ratio) cannot be computed exactly, because we do not have access to the behavioral policy $\pi_b(a|s)$. This means we cannot apply gradient-based MORL approaches (e.g., multiple-gradient descent or PCGrad), because we cannot calculate the gradient for this objective. We also cannot apply MO-MPO, because we cannot compute the temperature $\eta_2$ that satisfies the specified trade-off between objectives. In contrast, we can apply DiME and LS by simplifying the update rule, while still satisfying the specified trade-off. We described this in Section 5.2 and have added clarification in Appendix B.4.

---

### Author Response · Authors · 2021-11-22
**Author follow-up**

We would appreciate hearing back from the reviewers on whether their concerns have been addressed by our responses and updated paper. We are happy to answer any remaining questions. Thank you.

---

### Decision · Program_Chairs · 2022-01-20

**Decision:**

Reject

**Comment:**

The reviewers in general did not seem to be strongly impressed by the contribution of the paper. As the authors noted, some reviewers seemed to misinterpret the claims of the paper --- the paper is not to design new MORL algorithms that are significantly better on standard MORL benchmarks but is to apply MORL on offline RL and fine-tuning. On the other hand, the AC suspects that the paper's exposition could be more centered around the applications, e.g., arguing why offline RL can be benefited from better MO training, and why the challenge of offline RL is to balance some given notions of risk and return computationally (instead of, e.g., developing the right notion/formula for quantifying the risk and return.) Moreover, I think the paper would be stronger if the evaluation for offline RL setting can be made stronger, e.g., including more tasks and algorithms on the D4RL dataset. If the paper's claim is that MORL is a great tool for offline RL, perhaps it's useful to demonstrate that MORL can achieve SOTA reliably when used on top of existing offline RL algorithms (which almost always have two parts in the objective). In summary, in the AC's opinion, the paper has a valuable contribution to the community but is somewhat boardline for ICLR in the current form, and the AC encourages the authors to resubmit to a top venue conference after addressing some of the reviewers' comments.